# The manifold actions of signaling peptides on subcellular dynamics of a receptor specify stomatal cell fate

Xingyun Qi[1†], Akira Yoshinari[2], Pengfei Bai[3], Michal Maes[1‡], Scott M Zeng[3,4], Keiko U Torii[1,2,3*]

[1]Howard Hughes Medical Institute and Department of Biology, University of Washington, Seattle, United States; [2]Institute of Transformative Biomolecules (WPI-ITbM), Nagoya University, Aichi, Japan; [3]Howard Hughes Medical Institute and Department of Molecular Biosciences, The University of Texas at Austin, Austin, United States; [4]Department of Physics, University of Washington, Seattle, United States

**Abstract** Receptor endocytosis is important for signal activation, transduction, and deactivation. However, how a receptor interprets conflicting signals to adjust cellular output is not clearly understood. Using genetic, cell biological, and pharmacological approaches, we report here that ERECTA-LIKE1 (ERL1), the major receptor restricting plant stomatal differentiation, undergoes dynamic subcellular behaviors in response to different EPIDERMAL PATTERNING FACTOR (EPF) peptides. Activation of ERL1 by EPF1 induces rapid ERL1 internalization via multivesicular bodies/late endosomes to vacuolar degradation, whereas ERL1 constitutively internalizes in the absence of EPF1. The co-receptor, TOO MANY MOUTHS is essential for ERL1 internalization induced by EPF1 but not by EPFL6. The peptide antagonist, Stomagen, triggers retention of ERL1 in the endoplasmic reticulum, likely coupled with reduced endocytosis. In contrast, the dominant-negative ERL1 remained dysfunctional in ligand-induced subcellular trafficking. Our study elucidates that multiple related yet unique peptides specify cell fate by deploying the differential subcellular dynamics of a single receptor.

**\*For correspondence:**
ktorii@utexas.edu

**Present address:** [†]Department of Biology, Rutgers University, Camden, United States; [‡]Institute of Systems Biology, Seattle, United States

**Competing interests:** The authors declare that no competing interests exist.

## Introduction

Receptor-mediated endocytosis is an integral part of cellular signaling, as it mediates signal attenuation and provides spatial and temporal dimensions to signaling events. In mammalian systems, endocytosis of receptor tyrosine kinases can attenuate the signal outputs, by removing the active receptor pools from the plasma membrane, or it can specify signals at defined sites of action, such as signaling through endosomes (*Sigismund et al., 2012*). As a sessile organism, plants make use of a large number of receptor-like kinases (RLKs) for cell-cell, shoot-to-root, and inter-kingdom communications (*Shiu and Bleecker, 2001*). The RLKs with extracellular leucine-rich repeat domain, known as LRR-RLKs, comprise the largest RLK subfamily (*Shiu and Bleecker, 2001*), and they specify critical aspects of development, environmental response, and immunity by perceiving extrinsic signals (*Torii, 2004*; *Macho and Zipfel, 2014*). Increasing evidence shows that the subcellular localization and trafficking routes of LRR-RLKs regulate their function and activity (*Ben Khaled et al., 2015*). In Arabidopsis, bacterial flagellin peptide flg22 induces the heterodimer formation consisting of the LRR-RLKs FLAGELLIN SENSING2 (FLS2) and BRI1-ASSOCIATED RECEPTOR KINASE (BAK1)/SOMATIC EMBRYOGENESIS RECEPTOR-LIKE KINASE 3 (*Chinchilla et al., 2007*). This triggers the endocytosis and degradation of the receptor complex to generate transient cellular immune signaling but also to prevent continuous signaling to the same stimulus (*Robatzek et al., 2006*;

*Beck et al., 2012*). The brassinosteroid (BR) receptor BRASSINOSTEROID INSENSITIVE1 (BRI1) forms a complex with BAK1 (*Li et al., 2002*; *Nam and Li, 2002*; *Bücherl et al., 2013*). BRI1 can undergo constitutive endocytosis independent of BRs, but BRs can elevate BRI1 and BAK1 interaction and reduce the number of available BRI1-BAK1 complexes on the plasma membrane (*Geldner et al., 2007*; *Bücherl et al., 2013*; *Hutten et al., 2017*). CLAVATA1 (CLV1), an LRR-RLK that controls stem cell homeostasis within the shoot meristem (*Clark et al., 1997*), is downregulated by ligand-dependent internalization upon perception of its ligand CLV3 (*Nimchuk et al., 2011*). It remains a key question as to where within the cell these LRR-RLKs transduce signals and how different activation states of LRR-RLKs influence their subcellular localization.

Developmental patterning of stomata, adjustable pores on the plant epidermis for gas-exchange and transpiration, relies on intricate cell-cell communication mediated by signaling peptides and their receptors (*Lau and Bergmann, 2012*; *Pillitteri and Torii, 2012*). In Arabidopsis, secreted peptides from the EPF family, and their shared receptor LRR-RLKs, ERECTA, ERL1, and ERL2, mediate this process (*Rychel et al., 2010*). Amongst the plant LRR-RLKs, the ERECTA family offers a unique advantage to study how multiple signals are perceived to achieve cell fate and patterning. EPF2 and EPF1 negatively regulate stomatal development primarily through ERECTA and ERL1, respectively (*Hara et al., 2007*; *Hara et al., 2009*; *Hunt and Gray, 2009*). In contrast, EPF-LIKE9 (EPFL9), also known as Stomagen, promotes stomatal development by competing with EPF2 and, to some extent, with EPF1 for receptor binding (*Sugano et al., 2010*; *Lee et al., 2015*; *Lin et al., 2017*; *Qi et al., 2017*). Moreover, EPFL4/5/6, a subfamily only expressed in hypocotyls and stems, also act as ligands for the ERECTA family to inhibit stomatal formation when an LRR receptor protein, TOO MANY MOUTHS (TMM), is missing (*Abrash and Bergmann, 2010*; *Abrash et al., 2011*). Although the final phenotypic outcomes of these different EPF signaling events are well characterized, the very early step of signal transmission by the receptors remain elusive. While internalization of ERL2 was documented briefly (*Ho et al., 2016*), it is unknown whether it has any implications in signal transduction or in which subcellular organelle ERL2 was localized.

ERL1 regulates guard cell differentiation in an autocrine manner in addition to enforcing stomatal spacing of neighboring cells in a paracrine manner (*Lee et al., 2012*; *Qi et al., 2017*). This dual function of ERL1 can be attributed to its cell-type specific expression patterns as well as its ability to perceive different EPF/EPFL peptide ligands (*Shpak et al., 2005*; *Lin et al., 2017*). On the other hand, its sister receptor ERECTA is broadly and ubiquitously expressed in the epidermis (*Horst et al., 2015*). The specific expression of ERL1 in stomatal meristemoids as well as its dual function as an autocrine and paracrine signaling receptor provide a unique advantage to study how receptor subcellular dynamics translates into the eventual cell fate. Here, we combined genetic, pharmacological, and live imaging approaches to explore the initial events that occurred at ERL1 upon perception of different EPF peptides. Our study shows that EPF1 and EPFL6, the ligands activating the inhibitory stomatal signaling, trigger ERL1 endocytosis into multi-vesicular bodies/late endosomes (MVB/LEs). TMM, which can form a receptor complex with ERL1, is required for the EPF1-induced ERL1 internalization and suppression of stomatal fate but is superfluous for EFPL6-induced ERL1 internalization. Surprisingly, Stomagen interferes with the inhibitory regulation of stomatal differentiation by retaining ERL1 to the endoplasmic reticulum, similar to when endocytosis was pharmacologically blocked by Tyrphostin A23 (Tyr A23) (*Santuari et al., 2011*) and Endosidin 9–17 (ES9-17) (*Dejonghe et al., 2019*). Additionally, we extensively examined the effects of Brefeldin A (BFA) on subcellular organelle behaviors in stomatal meristemoids and established optimal conditions to observe BFA body formation in true leaves. Combined, our study reveals a mechanism by which plant cells interpret multiple signals through the subcellular localization and trafficking route of a single receptor.

## Results

### ERL1 undergoes endocytosis through multivesicular bodies to vacuole in stomatal meristemoids

To understand how stomatal cell fate decisions are made at the level of receptor subcellular dynamics, we first examined the localization of ERL1 (*Figure 1*). As reported previously (*Qi et al., 2017*), a functional ERL1-YFP fusion protein driven by its endogenous promoter (*ERL1pro::ERL1-YFP*) in *erl1* seedlings marks the plasma membrane of stomatal-lineage cells, most notably late meristemoids. In

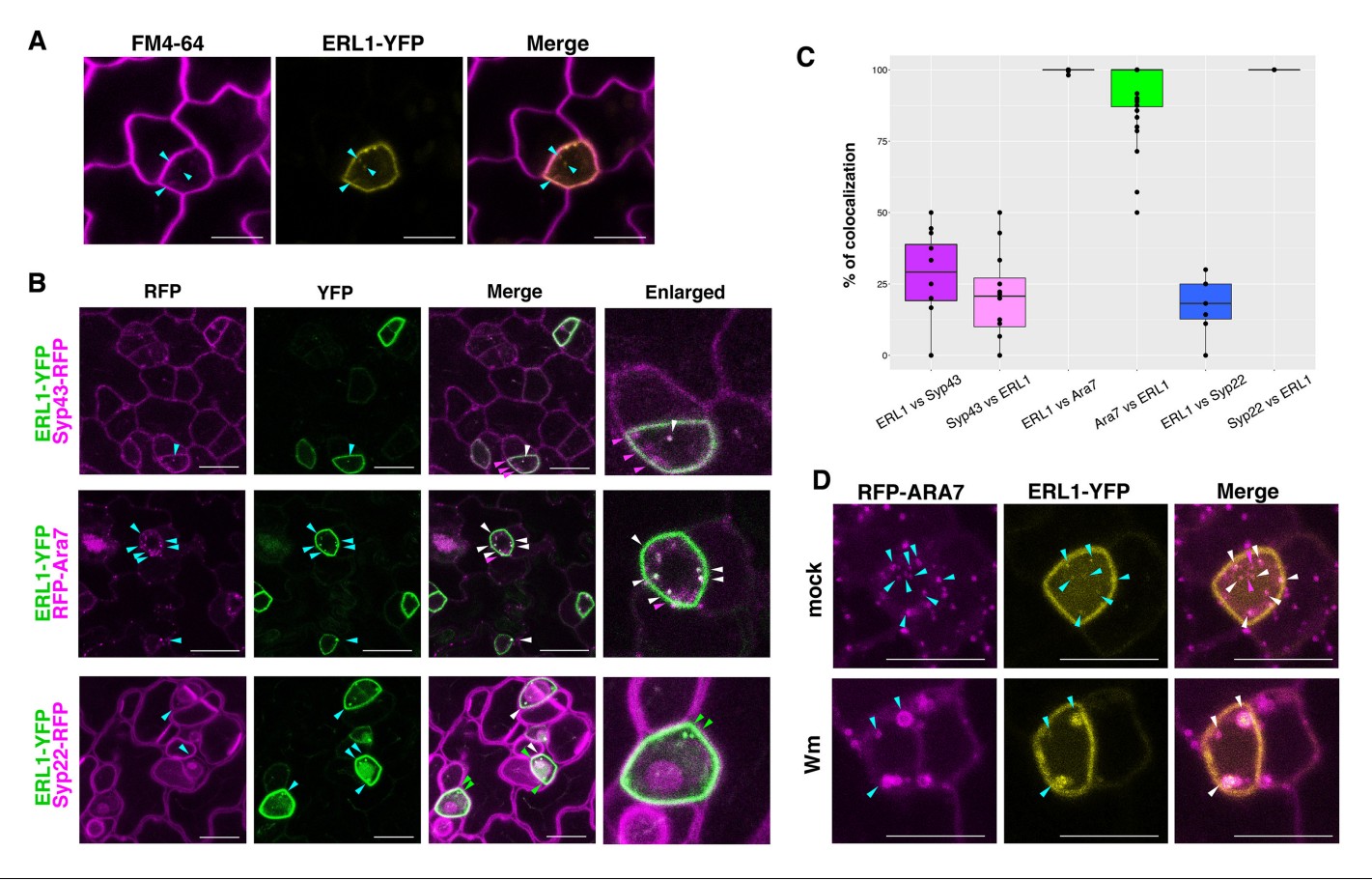

**Figure 1.** ERL1-YFP has dual localization on plasma membrane and late endosomes. (**A**) Representative confocal microscope images of ERL1-YFP expression in a meristemoid (yellow) co-stained with 10 μM FM4-64 (magenta). Cyan arrowheads, ERL1-YFP endosomes co-localizing with FM4-64 stains. Scale bars = 7.5 μm. (**B**) Representative confocal microscope images of ERL1-YFP co-localization analysis with the TGN marker SYP43-RFP (top), the MVB/LE marker RFP-ARA7 (middle), and the MVB/LE and vacuole marker SYP22-RFP (bottom) in the abaxial epidermis of developing true leaves of the 7-day-old seedlings. Merged images are shown in the third column, with enlarged images of representative meristemoids in the right column. Arrowheads point to endosomes bearing ERL1-YFP, SYP43-RFP, RFP-ARA7, and/or SYP22-RFP: cyan, single channels; green, YFP; magenta, RFP; white, co-localization . Scale bars = 10 μm. (**C**) Quantitative analysis of the co-localized endosomes between ERL1-YFP and the subcellular marker proteins. Percentage of the endosomes of the former protein that co-localize with the latter protein is shown as dots. Lines in the boxplot show the median value of each group, and the boxes represent from the first to third quartiles. Number of cells analyzed, n = 40 for ERL1 vs ARA7 or ARA7 vs ERL1; n = 12 for ERL1 vs SYP43 or SYP43 vs ERL1; n = 7 for ERL1 vs SYP22 or SYP22 vs ERL1. (**D**) ERL1-YFP and RFP-ARA7 treated with Wm. Shown are RFP-ARA7 (left column) and ERL1-YFP (middle column) in the abaxial epidermis of developing true leaves of the 7-day-old seedlings treated with mock (top row) or 30 μM Wm (bottom row). Arrowheads point to ERL1-YFP and/or RFP-ARA7 endosomes: cyan, single channels; magenta, YFP; white, co-localization. Scale bars = 10 μm.

The online version of this article includes the following figure supplement(s) for figure 1:

**Figure supplement 1.** RFP-ARA7 expressing endosomes and Wm bodies.

addition, we detected some punctae highlighted by ERL1-YFP that co-localized with FM4-64, a styryl dye used to trace the endocytic pathways in plants (*Meckel et al., 2004*; *Figure 1A*). To define the subcellular localization of ERL1-YFP, its co-localization analysis was performed with marker proteins SYP43-RFP for *trans*-Golgi network/early endosomes (TGN/EEs), RFP-ARA7 for MVB/LE, and SYP22-RFP for vacuoles (occasionally MVB/LE)(*Ebine et al., 2011*; *Postma et al., 2016*; *Figure 1B*, *Video 1*). ERL1-YFP extensively co-localizes and moves together with RFP-ARA7 (*Figure 1B,C*, *Video 1*), whereas only 25% and 18% of ERL1-YFP-positive punctae are also labelled by SYP43-RFP and SYP22-RFP, respectively. Thus, ERL1-YFP predominantly resides on the MVB/LE. This is further confirmed by a pharmacological approach using Wortmannin (Wm), a fungal drug that can cause fusion of MVB/LEs by inhibiting phosphatidylinositol-3 (PI3) and phosphatidylinositol-4 (PI4) kinases

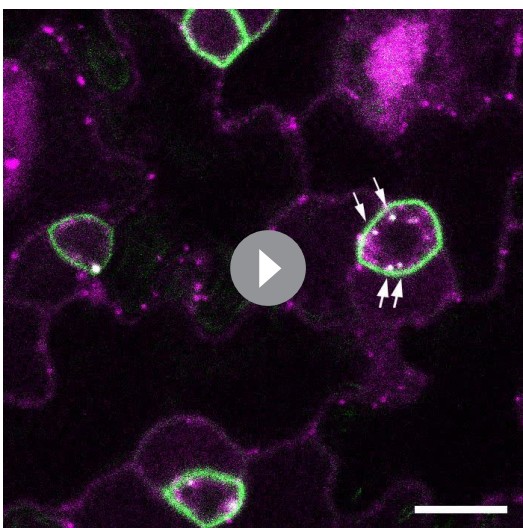

**Video 1.** ERL1-YFP and RFP-Ara7. A time-lapse movie of an Arabidopsis meristemoid co-expressing ERL1-YFP and RFP-Ara7. Images were taken every 30 s for 10 frames and presented as four frames per second. https://elifesciences.org/articles/58097#video1

(*Foissner et al., 2016*). The Wm application on Arabidopsis seedlings resulted in the formation of typical ring-like Wm bodies marked by both ERL1-YFP and RFP-ARA7 (*Figure 1D*; *Figure 1—figure supplement 1A*). These Wm bodies are much larger (average size of 0.993 ± 0.203 μm) than the endosomes (average size of 0.383 ± 0.063 μm), thus easy to recognize and quantify (*Figure 1—figure supplement 1B*).

Receptors internalized via MVB/LE are destined for degradation in lytic vacuoles (*Reyes et al., 2011*). Because we did not observe clear accumulation of ERL1 in a vacuole, and co-localization of SYP22-RFP and ERL1-YFP was limited to the endosomes (*Figure 1B,C*), we took a pharmacological approach using Concanamycin A, a specific inhibitor of vacuolar H$^+$-ATPase that is known to reduce protein degradation in the lytic vacuole (*Kleine-Vehn et al., 2008*). As shown in *Figure 2*, Concanamycin A treatment following the wash-out of the endocytic tracer dye FM4-64 led to accumulation of strong vacuolar YFP signals surrounded by an FM4-64 stained tonoplast. Taken together, our results indicate that, within the stomatal precursor cells, ERL1 undergoes endocytic trafficking from the plasma membrane to MVB/LE and is destined to vacuolar degradation.

## The effects of BFA on endomembrane behaviors in stomatal meristemoids

Endocytosis is an essential process to regulate cell signaling by controlling the turnover of plasma membrane proteome. Activated plant receptor kinases can either be recycled back to the plasma membrane or are destined for endocytic degradation via MVB/LE for signal termination. BFA, a chemical inhibitor of ADP-ribosylation factor guanine-nucleotide exchange factors (ARF-GEFs) including GNOM has been widely used to block endosomal recycling and endocytosis of membrane proteins (*Geldner et al., 2003*). BFA treatment in roots results in formation of characteristic BFA bodies, which are conglomerates of TGN surrounded by the Golgi apparatus (*Geldner et al., 2003*). It has been reported, however, that BFA application to Arabidopsis leaves triggers re-absorption of Golgi membrane collapsed onto the endoplasmic reticulum (*Robinson et al., 2008*; *Langhans et al., 2011*). Thus, to study the effects of BFA on ERL1 endomembrane dynamics, we first sought to fully document the BFA effects in stomatal meristemoids. The previous reports used high concentration (e.g. 50, and 90, 180, 360 μM of BFA) (*Robinson et al., 2008*; *Langhans et al., 2011*) compared to the lower concentration (30 μM of BFA) generally used in roots.

First, we treated Arabidopsis seedlings expressing the endoplasmic reticulum marker, GFP-HDEL (*Mitsuhashi et al., 2000*) or the Golgi marker ST-YFP (*Takagi et al., 2013*) co-stained with FM4-64 with different BFA concentrations (0, 30, 90, 180 μM) (*Figure 3*). Low BFA concentration at 30 μM does not confer any discernable effects on the endoplasmic reticulum (GFP-HDEL; *Figure 3A,B*). Under this condition, BFA bodies (magenta arrows) are surrounded by ST-YFP (green arrows, *Figure 3C*), exhibiting characteristic membrane organization of BFA bodies reported in roots (*Figure 3D*). These BFA bodies are also observed at 90 μM treatment (*Figure 3B,C*, magenta arrows). In contrast, BFA at very high concentration, 180 μM, conferred aberrant spherical structures of the endoplasmic reticulum (*Figure 3A,B*), which was also previously reported (*Nakano et al., 2009*). Indeed, at 180 μM BFA treatment, we observed Golgi absorbed ER, which confers ring-like structure of membrane accumulating ST-YFP (*Figure 3C*). These results demonstrate that previously reported effects of BFA on leaves were observed only when extremely high concentration of BFA was applied.

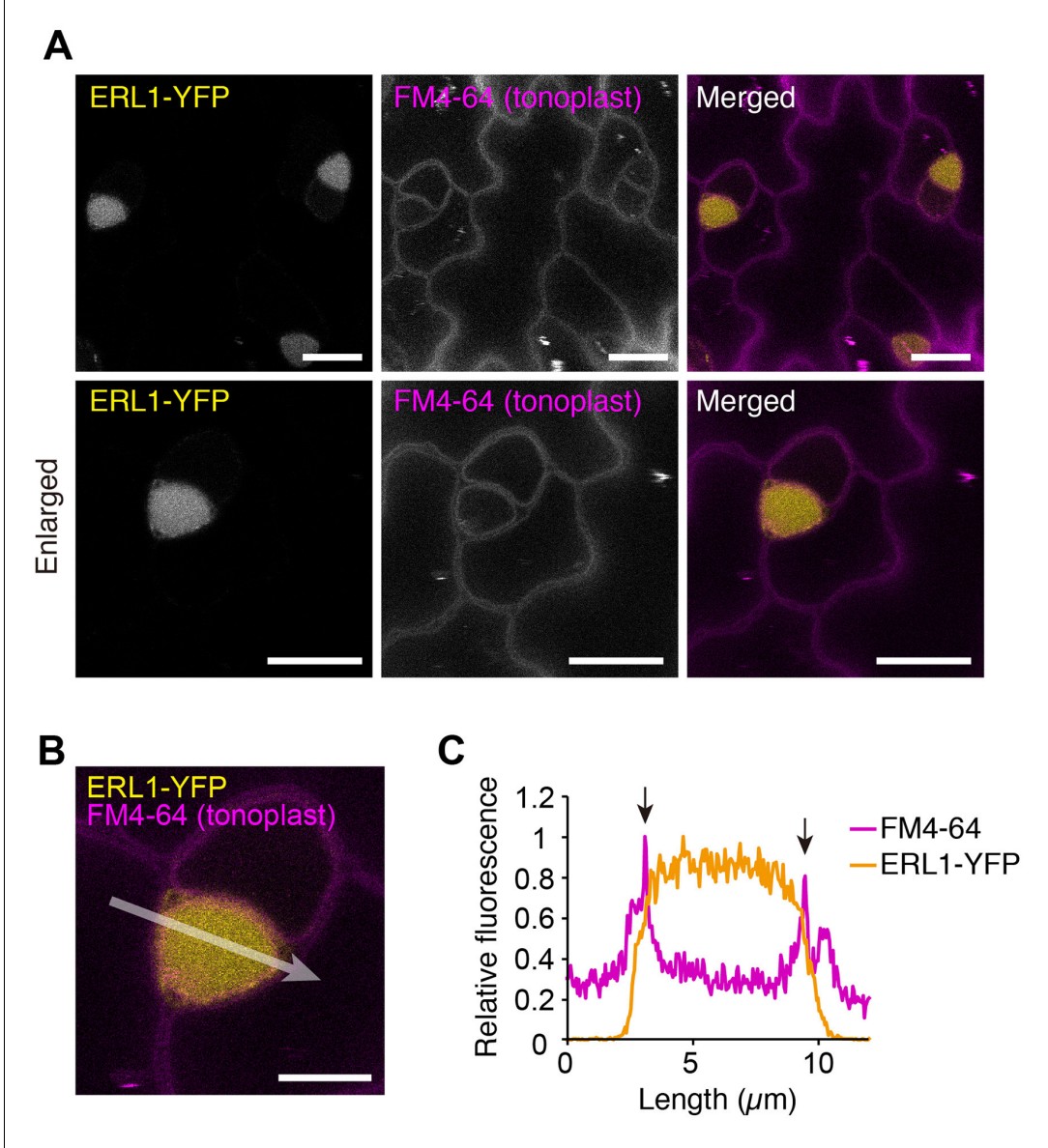

**Figure 2.** ERL1 is transported into vacuole. (**A**) Representative confocal microscopy images of ERL1-YFP (yellow) and FM4-64 (magenta) in the abaxial epidermis of developing true leaves of the 7-day-old seedlings expressing *ERL1pro::ERL1-YFP* in *erl1*. The seedlings were treated with 2 µM FM4-64 for 30 min followed by incubation in water for 6 hr and subsequently incubated in 1 µM Concancmycin A for 5 hr. Tonoplast is highlighted by FM4-64. Scale bars = 10 µm. (**B**) Enlarged merged image of ERL1-YFP (yellow) and FM4-64 (tonoplast, magenta). Scale bars = 5 µm. (**C**) Line plots of ERL1-YFP (yellow) and FM4-64 (magenta) fluorescence on 10-pixel-thickness of arrow indicated in (**B**). Arrows indicate FM4-64 fluorescence staining the tonoplast.

Next, to evaluate the intactness of different endomembrane organelles in stomatal meristemoids when treated with 30 µM BFA, we examined seven endomembrane markers, YFP-SYP32 (*cis*-Golgi), N-ST-YFP (*trans*-Golgi), GFP-SYP43 (TGN/EE), YFP-RabA1e (TGN/EE, plasma membrane), YFP-RabA5d (uncharacterized endosomes), ARA6-GFP (MVB/LE, plasma membrane) and YFP-ARA7 (MVB/LE) co-stained with FM4-64 (*Geldner et al., 2009*; *Ebine et al., 2011*; *Postma et al., 2016*). All marker lines show the formation of characteristic BFA bodies in meristemoids, without discernible collapse of Golgi apparatus into the endoplasmic reticulum (*Figure 3—figure supplement 1*). Taken together, our extensive documentation shows that 30 µM BFA confers BFA body formation just like when applied to roots. Therefore, BFA can be used to study the endocytosis of ERL1 in meristemoids.

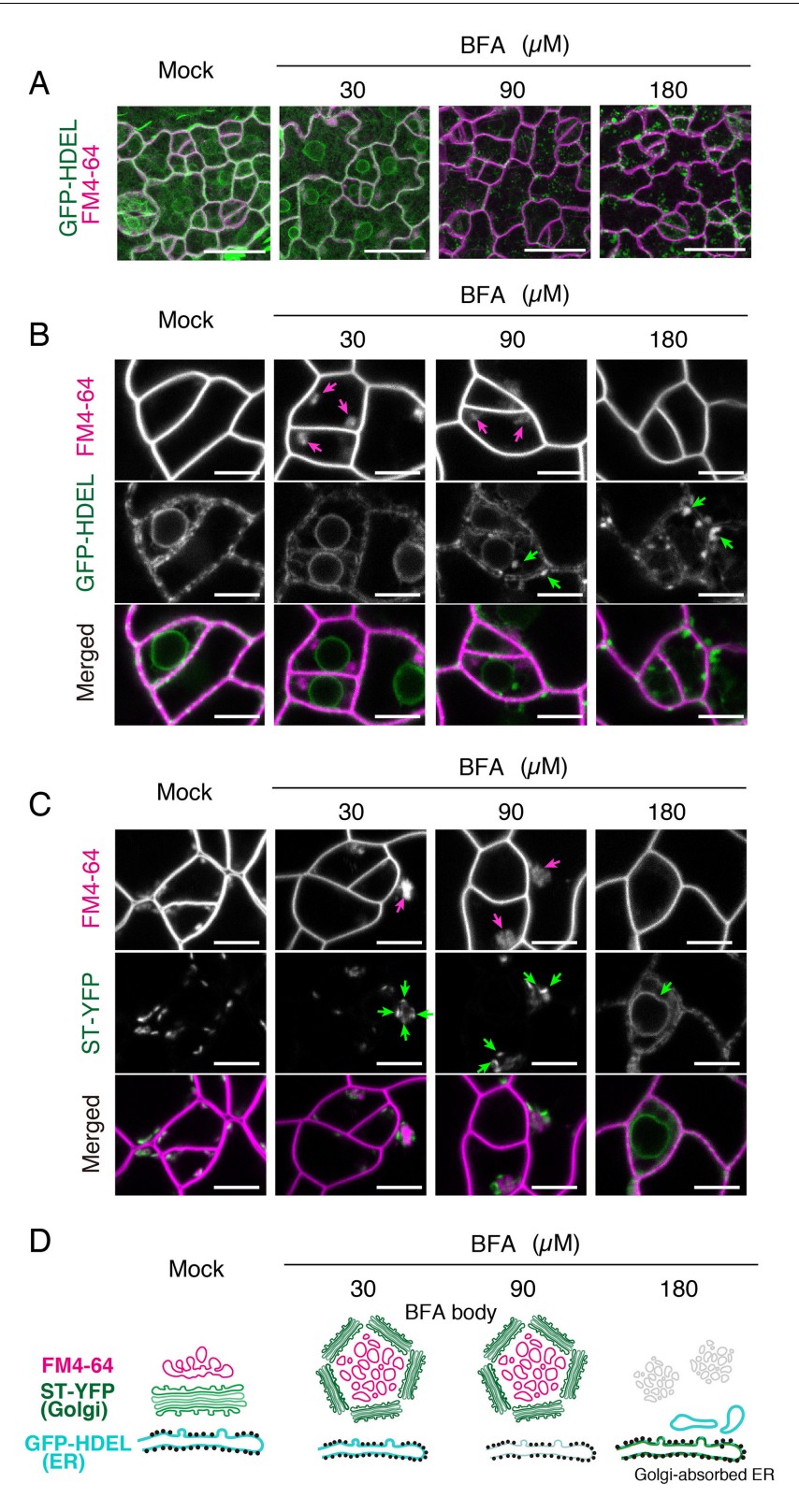

**Figure 3.** Effects of BFA on subcellular membrane structures in the meristemoids. (**A**) Representative Z-stack confocal images of an ER marker, GFP-HDEL (green) and FM4-64 (magenta) in true leaf abaxial epidermis from the 7-day-old seedlings expressing *CaMV35Spro::GFP-HDEL* treated with 5 µM FM4-64 alone (mock) or co-treated with 30, 90, 180 µM BFA for 1 hr. Scale bars = 20 µm. BFA treatment at low concentration (30 µM) does not alter the characteristic, mesh-like ER structure inside the meristemoids and at the edge of nuclei. In contrast, high concentration of BFA results in aberrant

*Figure 3 continued on next page*

*Figure 3 continued*

spherical structures in the ER. (**B**) Representative confocal images of an ER marker, GFP-HDEL (green) and FM4-64 (magenta) treated with BFA as described in (**A**). BFA treatment at low concentration (30 µM) causes the formation of BFA bodies (magenta arrows) without impacting the ER structure. In high BFA concentration confers aberrant spherical ER structure (green arrows). In addition, the FM4-64 signals in the BFA bodies disappear. Scale bars = 5 µm. (**C**) Representative confocal images of a Golgi marker, ST-YFP (green) and FM4-64 (magenta) in true leaf abaxial epidermis from the 7-day-old seedlings expressing *CaMV35Spro::N-ST-YFP* treated with 5 µM FM4-64 alone (mock) or co-treated with 30, 90, 180 µM BFA for 1 hr. Scale bars = 5 µm. In a lower and medium concentration (30 and 90 µM) of BFA, Golgi (green arrows) are surrounding BFA bodies (magenta arrows). By contrast, in higher concentration (180 µM), the Golgi marker becomes collapsed (green arrow) to ER. (**D**) Schematic diagrams depicting probable subcellular membrane structures in the meristemoids based on confocal microscopy.

The online version of this article includes the following figure supplement(s) for figure 3:

**Figure supplement 1.** Effect of BFA on Golgi apparatus and endosomal markers at a low concentration.

## TMM is required for the process of ERL1 endocytosis in true leaves

We wondered if the ERL1 endocytosis that we observed (*Figures 1* and *2*) is related to its biological signaling. A previous work has shown that ERL1 forms a heterodimer with TMM, a receptor protein, to create a pocket for the proper binding of its major ligand EPF1 (*Lee et al., 2012*; *Lin et al., 2017*). The absence of TMM results in clustered stomata (*Figure 4A*), indicating that TMM is required for EPF1-ERL1 signaling to enforce proper stomatal spacing (*Hara et al., 2007*; *Lee et al., 2012*). As a first step to test whether active ERL1 signaling is a prerequisite for its endocytosis, we monitored ERL1 dynamics in *tmm* background (*Figure 4*). The number of ERL1-YFP-positive endosomes per meristemoid cell was reduced in *tmm* (*Figure 4B*; p=1.58e-14, *Figure 4—figure supplement 1*). Fluorescent volume intensity (voxel) ratio analysis also revealed statistically-significant reduction of ERL1-YFP-positive endosomes in *tmm* (*Figure 4D*, p=0.0049) as well as significant reduction of ERL1-YFP-positive BFA bodies in 30 µM BFA-treated *tmm* (*Figure 4D*, p=4.64e-11) compared to those in wild type. To assess the role of the secretory pathway in the observed difference in ERL1-YFP BFA bodies between wild-type and *tmm*, the BFA treatment was further performed in the presence of protein synthesis inhibitor cycloheximide (CHX). Treatment with 50 µM CHX for 1 hr followed by either mock or 30 µM BFA led to reduction in the number of ERL1-YFP-positive BFA bodies in *tmm* mutant (*Figure 4—figure supplement 1*, p=0.0154). Thus, ERL1-YFP-positive BFA bodies are reduced in *tmm* mutant regardless of the presence or absence of CHX, thereby suggesting that the absence of TMM does not impact the ERL1 secretory pathway. Next, we treated the seedlings with Wm, which conferred significant reduction of ERL1-YFP-labelled Wm-bodies in *tmm* compared to that in wild type (*Figure 4E,F*, p=1.40e-5). Combined, the results suggest that TMM is critical for the endocytosis/internalization of ERL1 to MVB/LE.

To rule out the possibility that TMM influences a general endocytic degradation machinery, we examined the effects of *tmm* on general endocytosis using FM4-64. In wild type, 92.8% (n = 20 cells) of ERL1-YFP-labelled endosomes can be stained by FM4-64. In *tmm*, however, FM4-64 still internalizes to endosomes whereas ERL1-YFP internalization can only be detected in 30% of the cells examined (n = 30 cells) (*Figure 4—figure supplement 2A*). The effects of *tmm* on the formation of MVB/LEs were subsequently tested (*Figure 4—figure supplement 2B–E*). In the cells co-expressing RFP-ARA7 and ERL1-YFP, no significant difference was observed in the numbers of RFP-ARA7-labelled endosomes and Wm bodies per meristemoid cell between wild type and *tmm* (*Figure 4—figure supplement 2B,C*). In contrast, the *tmm* mutation conferred reduction in ERL1-YFP-labelled Wm bodies that co-localized with RFP-ARA7 (*Figure 4—figure supplement 2B–C*). Thus, the results suggest that TMM mediates endocytosis of ERL1 for vacuolar sorting without influencing general endomembrane trafficking.

To further explore the role of TMM for the ERL1 receptor dynamics, we performed fluorescence recovery after photobleaching (FRAP) assays on ERL1-YFP on the plasma membrane (*Figure 4—figure supplement 3*), and the half time of fluorescence recovery was calculated from modeling to exponential curves (*Figure 4G*. H, *Videos 2* and *3*). In wild type, the calculated mean half-time of ERL1-YFP fluorescence recovery (t1/2) was 23.55 ± 5.55 s, whereas in *tmm* it was 70.89 ± 24.63 s (*Figure 4G*. H). The longer recovery time of ERL1-YFP in *tmm* could be due to reduced diffusion rate/mobility of ERL1 receptor at the plasma membrane or, alternatively, slower cycle of ERL1 receptor between the plasma membrane and the cytoplasmic fractions. The results imply that, in the

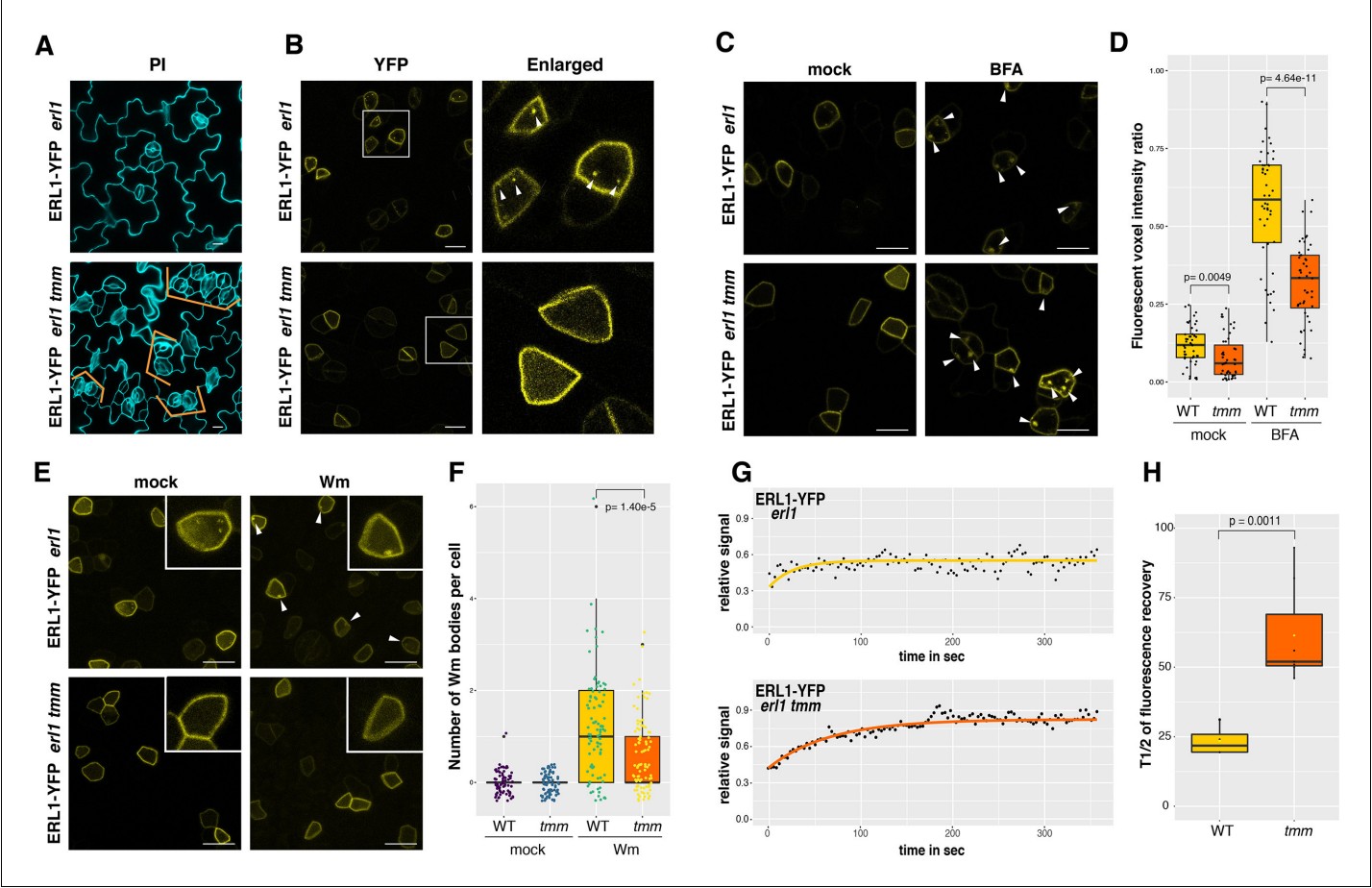

**Figure 4.** ERL1 internalization requires its co-receptor TMM. (**A**) Representative confocal microscopy images of PI-stained true leaf abaxial epidermis of ERL1-YFP in *erl1* (top) and in *erl1 tmm* (bottom), the latter shows characteristic stomatal cluster phenotype (orange brackets). Scale bars = 10 μm. (**B**) ERL1-YFP subcellular localization in *erl1* (top row) and in *erl1 tmm* (bottom row) in the true leaf abaxial epidermis from the 7-day-old seedlings. Right column; enlarged images. Their stomatal phenotypes are shown in (**A**). Arrowheads indicate endosomes. Scale bars = 10 μm. (**C**) Representative confocal images of ERL1-YFP in *erl1* (top row) or in *erl1 tmm* (bottom row) of the abaxial epidermis of developing true leaves from the 7-day-old seedlings treated with mock (left column) or 30 μM BFA (right column). Arrowheads indicate BFA bodies. Scale bars = 10 μm. (**D**) Quantitative analysis of the volume ratio of YFP-positive endosomes and BAF bodies per cell volume (voxels) when ERL1-YFP in *erl1* (yellow) or *erl1 tmm* (orange) are treated with mock or 30 μM BFA. Welch's Two sample T-test was performed for pairwise comparisons. Experiments were repeated three times. Two-way ANOVA analysis: genotype (WT, *tmm*), p<3.85e-14; treatment (mock, BFA), p<2.2e-16. The total numbers of cells subjected to YFP signal intensity measurements are 86, 130, 66 (WT mock); 138, 94, 131 (*tmm* mock); 230, 336, 109 (WT BFA); 300, 393, 211 (*tmm* BFA). (**E**) Representative images of ERL1-YFP in *erl1* (top row) or in *erl1 tmm* (bottom row) treated with mock (left column) or 25 μM Wm (right column). Inset, an enlarged image of a representative meristemoid. Arrowheads indicate Wm bodies. Scale bars = 10 μm. (**F**) Quantitative analysis of the number of Wm bodies per cell when ERL1-YFP in *erl1* (yellow) or *erl1 tmm* (orange) are treated with mock or 25 μM Wm. Lines in the boxplot show the median value, and each dot represents individual data point with jitter. Welch's Two sample T-test was performed for pairwise comparisons. Two-way ANOVA analysis: genotype (WT, *tmm*), p=1.597e-6; treatment (mock, Wm), p<2.2e-16. Experiments were repeated three times. the total numbers of cells counted are 74 (WT mock); 91 (*tmm* mock); 82 (WT Wm); 90 (*tmm* Wm). (**G**) FRAP analyses of plasma membrane ERL1-YFP in wild type (*erl1*) or in *tmm* (*erl1 tmm*). Shown are representative fluorescence recovery curves plotted as a function of time and fitted to Single Exponential Fitting. ERL1-YFP in *erl1* (top; yellow); ERL1-YFP in *erl1 tmm* (bottom; orange). (**H**) Quantitative analysis of the half time of fluorescence recovery of plasma membrane ERL1-YFP in *erl1* (yellow) and *erl1 tmm* (orange). Lines in the boxplot show the median value. T-test was performed for pairwise comparisons between *erl1* and *erl1 tmm*. n = 3 for WT and n = 9 for *tmm*.

The online version of this article includes the following figure supplement(s) for figure 4:

**Figure supplement 1.** ERL1 BFA body formation in wild-type, *tmm*, or dominant-negative ERL1 background in the presence of cycloheximide.

**Figure supplement 2.** *tmm* mutation does not affect general internalization and the MVB structure.

**Figure supplement 3.** Observed fluorescence intensity change during FRAP experiments.

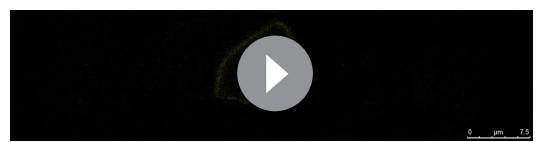

**Video 2.** FRAP analysis of ERL1-YFP in *erl1*. A time-lapse FRAP movie of ERL1-YFP in a meristemoid in *erl1* mutant. 5 images were taken every 0.453 s (minimum speed) both before and during photobleaching, and then 120 frames were taken every 3 s after photobleaching. The movie was presented as five frames per second.
https://elifesciences.org/articles/58097#video2

absence of TMM, un-activated ERL1 receptors may not be readily targeted for the endocytic pathway and, consequently, remain stable on the plasma membrane.

## EPF1 triggers TMM-dependent ERL1 internalization

Of the 11 EPF family members, EPF1 is the major ligand for ERL1 (*Lee et al., 2012*). EPF1 signaling plays a negative role in stomatal development, and the induction of EPF1 peptide (iEPF1) confers arrested stomatal precursors (*Figure 5A*; *Hara et al., 2007*; *Lee et al., 2012*; *Qi et al., 2017*). We therefore tested whether the ERL1 internalization is ligand dependent. For this purpose, we first examined ERL1-YFP dynamics in *epf1* mutants. As shown in *Figure 5—figure supplement 1A*, both plasma membrane and highly mobile endosomes are highlighted by ERL1-YFP in *epf1*. When treated with BFA or Wm, the number of ERL1-YFP-positive BFA or Wm bodies per cell are similar between wild type and *epf1* (*Figure 5—figure supplement 1B–E*, p=0.512 and p=0.647 for BFA and Wm treatment, respectively). Therefore, although EPF1 is a primary ligand for ERL1, ERL1-YFP endocytosis is not severely affected in the absence of EPF1.

Considering the high similarity among the 11 EPF members, it is possible that the functional redundancy of other EPFs alleviates the defect of ERL1 internalization in *epf1*. To address this, we took advantage of the biologically-active mature EPF1 (MEPF1) peptide (*Figure 5*; *Lee et al., 2012*; *Qi et al., 2017*). Different concentrations of MEPF1 were applied to the true leaf epidermis of 7-day-old seedlings expressing *ERL1pro::ERL1-YFP*. The number of ERL1-YFP-positive endosomes per cell increases as the peptide concentration increases (Pearson correlation coefficient, r = 0.56, p=2.2 e-16; *Figure 5C,E*), indicating that MEPF1 peptide triggers the internalization of ERL1 in a dosage-dependent manner.

In the *tmm* background, however, the number of ERL1-YFP-positive endosomes per cell remains low regardless of the MEPF1 dosage applied (*Figure 5D,F*). Thus, in the absence of TMM, ERL1-YFP endocytosis is insensitive to MEPF1 application, consistent with the genetic evidence that the *tmm* mutation is epistatic to induced *EPF1* overexpression (*iEPF1*) (*Hara et al., 2007*; *Lee et al., 2012*; *Figure 5B*). Taken together, we conclude that EPF1 peptide ligand perception triggers the internalization of ERL1 receptor in a TMM-dependent manner.

## EPFL6 triggers ERL1 internalization in the absence of TMM

A previous structural analysis has shown that binding of EPF1 to the ERL1-TMM receptor complex does not lead to conformational change (*Lin et al., 2017*). To test if the pre-formed ERL1-TMM receptor complex is required for the internalization of ERL1, we took advantage of EPFL6, a peptide related to EPF1 with a distinct property (*Abrash and Bergmann, 2010*; *Abrash et al., 2011*; *Figure 6*). EPFL6 is normally expressed in the internal tissues of hypocotyls and stems, but not in the stomatal-lineage cells. Unlike EPF1, ectopic EPFL6 is a potent inhibitor of stomatal development even in the *tmm* mutant background (*Abrash and Bergmann, 2010*; *Abrash et al., 2011*; *Uchida et al., 2012*; *Figure 6A,B*). Using a similar strategy as MEPF1, we purified biologically active, predicted mature EPFL6 (MEPFL6) peptide. Indeed, the inhibition of stomatal formation by MEPFL6 is more sensitive in *tmm* mutant than in wild type

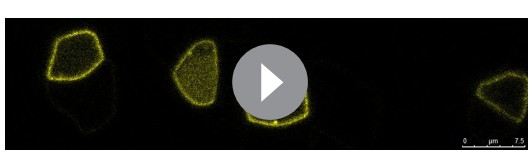

**Video 3.** FRAP analysis of ERL1-YFP in *erl1 tmm*. A time-lapse FRAP movie of ERL1-YFP in a meristemoid in *erl1 tmm* mutant. 5 images were taken every 0.453 s (minimum speed) both before and during photobleaching, and then 120 frames were taken every 3 s after photobleaching. The movie was presented as five frames per second.
https://elifesciences.org/articles/58097#video3

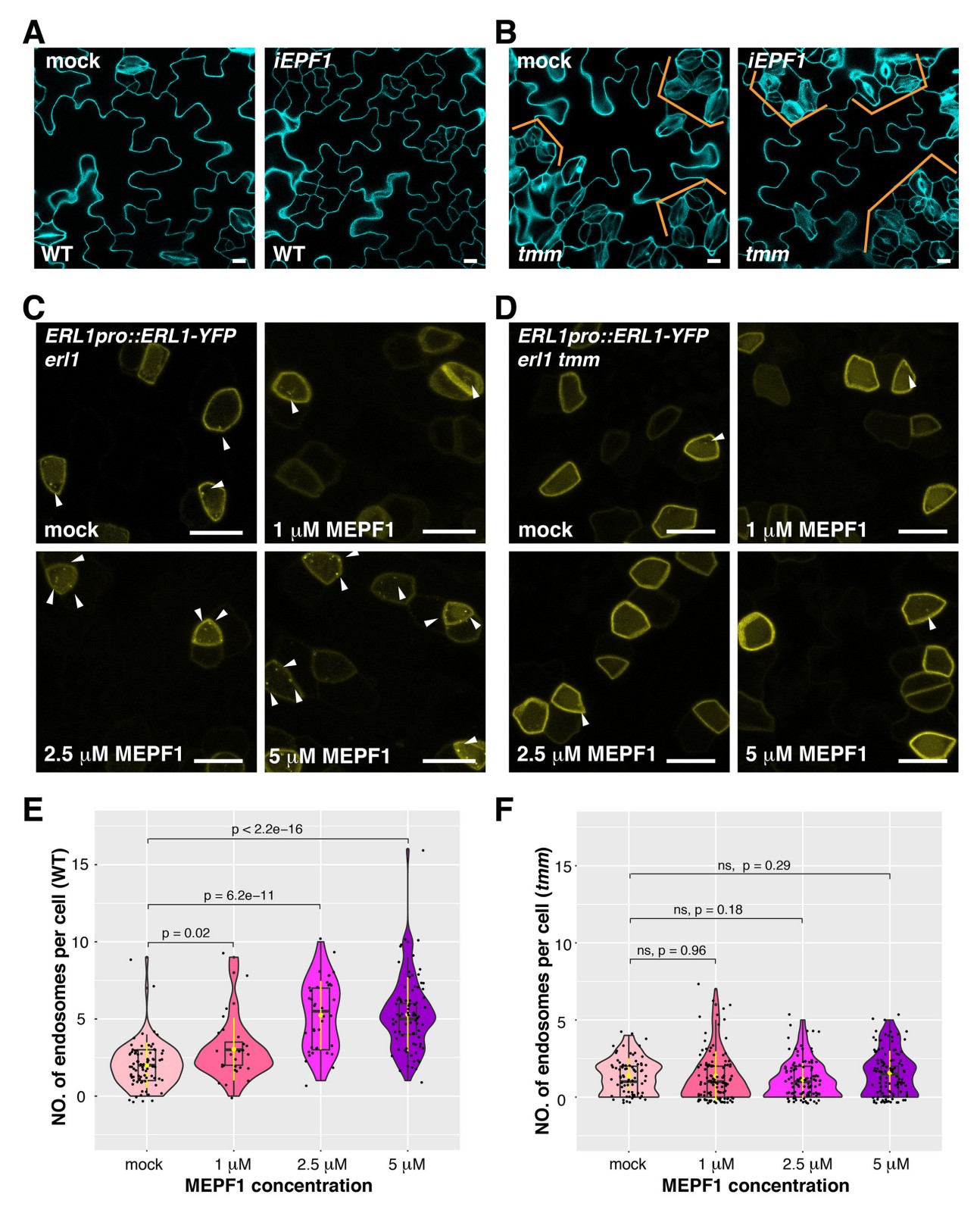

**Figure 5.** MEPF1 triggers ERL1-YFP internalization in *erl1* but not in *erl1 tmm*. (**A**) Representative confocal microscopy images of cotyledon abaxial epidermis from the 4-day-old *iEPF1* seedlings treated with mock (left) or 10 μM Estradiol (right). Scale bars = 10 μm. (**B**) Representative confocal microscopy images of cotyledon abaxial epidermis from the 4-day-old *iEPF1* in *tmm* seedlings treated with mock (left) or 10 μM Estradiol (right). Brackets indicate clustered stomata in both mock- and estradiol-induced samples. Scale bars = 10 μm. (**C**) Representative confocal microscopy images

*Figure 5 continued on next page*

Figure 5 continued

of ERL1-YFP in *erl1* treated with mock (top left), 1 μM MEPF1 (top right), 2.5 μM MEPF1 (bottom left) and 5 μM MEPF1 (bottom right) are shown. Arrowheads indicate endosomes. Scale bars = 10 μm. (D) Representative confocal microscopy images of ERL1-YFP in *erl1 tmm* treated with mock (top left), 1 μM MEPF1 (top right), 2.5 μM MEPF1 (bottom left) and 5 μM MEPF1 (bottom right) are shown. Arrowheads indicate endosomes. Scale bars = 10 μm. (E) Quantitative analysis of the number of ERL1-YFP-positive endosomes per cell at different concentrations of MEPF1 application in *erl1* shown as a violin plot. Dots, individual data points. Median values are shown as lines in the boxplot, and mean values are shown as yellow dots in the plot. Welch's two sample T-test was performed for pairwise comparisons of samples treated with the mock and different concentration of MEPF1. Number of cells analyzed, n = 79, 27, 38, 82 for treatment with mock, 1 μM, 2.5 μM, 5 μM MEPF1. (F) Quantitative analysis of the number of ERL1-YFP-positive endosomes per cell at different concentrations of MEPF1 application in *erl1 tmm* shown as a violin plot. Dots, individual data points. Median values are shown as lines in the boxplot, and mean values are shown as yellow dots in the plot. Welch's two sample T-test was performed for pairwise comparisons of samples treated with the mock and different concentration of MEPF1. Number of cells analyzed, n = 76, 113, 109, 114 for treatment with mock, 1 μM, 2.5 μM, and 5 μM MEPF1, respectively.

The online version of this article includes the following figure supplement(s) for figure 5:

**Figure supplement 1.** Absence of endogenous EPF1 does not affect ERL1-YFP internalization.

(*Figure 6—figure supplement 1*). In contrast to MEPF1, MEPFL6 application induced ERL1-YFP internalization in a dosage-dependent manner regardless of the presence or absence of TMM (*Figure 6C–F*). The results indicate that TMM is not required for EPFL6-triggered ERL1-YFP internalization. Rather, the ERL1-YFP endocytosis accurately reflects the activity of ERL1 signaling to inhibit stomatal development (*Figure 6* and *Figure 6—figure supplement 1*), thereby supporting the notion that distinct EPF/EPFL peptide ligands activate a sub-population of ERL1 receptor complexes to internalize through a TMM-based discriminatory mechanism.

## An antagonistic EPFL peptide, Stomagen, elicits retention of ERL1-YFP in the endoplasmic reticulum

Stomagen promotes stomatal development by competing with other EPFs for binding to the same receptor complex, including ERL1 (*Figure 7A*; *Kondo et al., 2010*; *Sugano et al., 2010*; *Lee et al., 2015*; *Lin et al., 2017*; *Qi et al., 2017*). Because the activated ERL1 receptor undergoes endocytosis to MVB/LEs, we sought to address the role of Stomagen on subcellular dynamics of ERL1. For this purpose, we first applied bioactive Stomagen peptide on seedlings expressing *ERL1pro::ERL1-YFP* in *erl1*. Unlike in mock-treated samples, YFP signal was detected inside the cells (*Figure 7—figure supplement 1A*). We subsequently treated Stomagen peptides to ERL1-YFP in the *erecta erl1 erl2* triple mutant background to remove any potential redundancy among the three ERECTA-family receptors. Strikingly, strong ERL1-YFP signals were detected in a ring-like structure surrounding the nucleus (*Figure 7B*), which co-localizes with the endoplasmic reticulum marker protein RFP-KDEL (*Faraco et al., 2017*; *Figure 7B*).

Next, to reveal a consequence of inactive ERL1 receptor on its subcellular dynamics, we applied Stomagen peptide on *tmm* seedlings expressing *ERL1pro::ERL1-YFP* and carefully reexamined the signal. Very faint ring-like structures were highlighted by ERL1-YFP in both mock and Stomagen-treated meristemoids (*Figure 7—figure supplement 1A*). This was enhanced in the *erecta erl1 erl2 tmm* quadruple mutant (*Figure 7C*). These ERL1-YFP signals co-localized with Rhodamine B hexyl esters, a dye that stains the endoplasmic reticulum (*Hawes et al., 2018*), suggesting that the absence of TMM intensifies accumulation of ERL1 in the endoplasmic reticulum (*Figure 7C*). To address the specificity of Stomagen effects on ERL1 subcellular behaviors, we applied Stomagen peptide to Arabidopsis seedlings expressing *BRI1pro::BRI1-GFP*, which is endogenously expressed in cotyledon and leaf epidermis (*Friedrichsen et al., 2000*). Strong BRI1-GFP signals are detected at the plasma membrane of stomatal meristemoids, guard cells, as well as pavement cells (*Figure 7—figure supplement 2*). Application of Stomagen peptide did not influence plasma membrane localization of BRI1-GFP (*Figure 7—figure supplement 2*), thus the endoplasmic reticulum accumulation of ERL1-YFP upon Stomagen perception is not a universal phenomenon to LRR-RLKs.

To biochemically characterize the effects of Stomagen application and *tmm* mutation on ERL1 accumulation in the endoplasmic reticulum, we further performed endoglycosidase H (Endo-H) enzymatic sensitivity assays. Endo-H cleaves N-glycans of proteins specifically in the endoplasmic reticulum, including LRR-RLKs (*Jin et al., 2007*; *Nekrasov et al., 2009*), but not the remodeled glycan chains of proteins transported to the Golgi or further. To detect slight molecular mass changes,

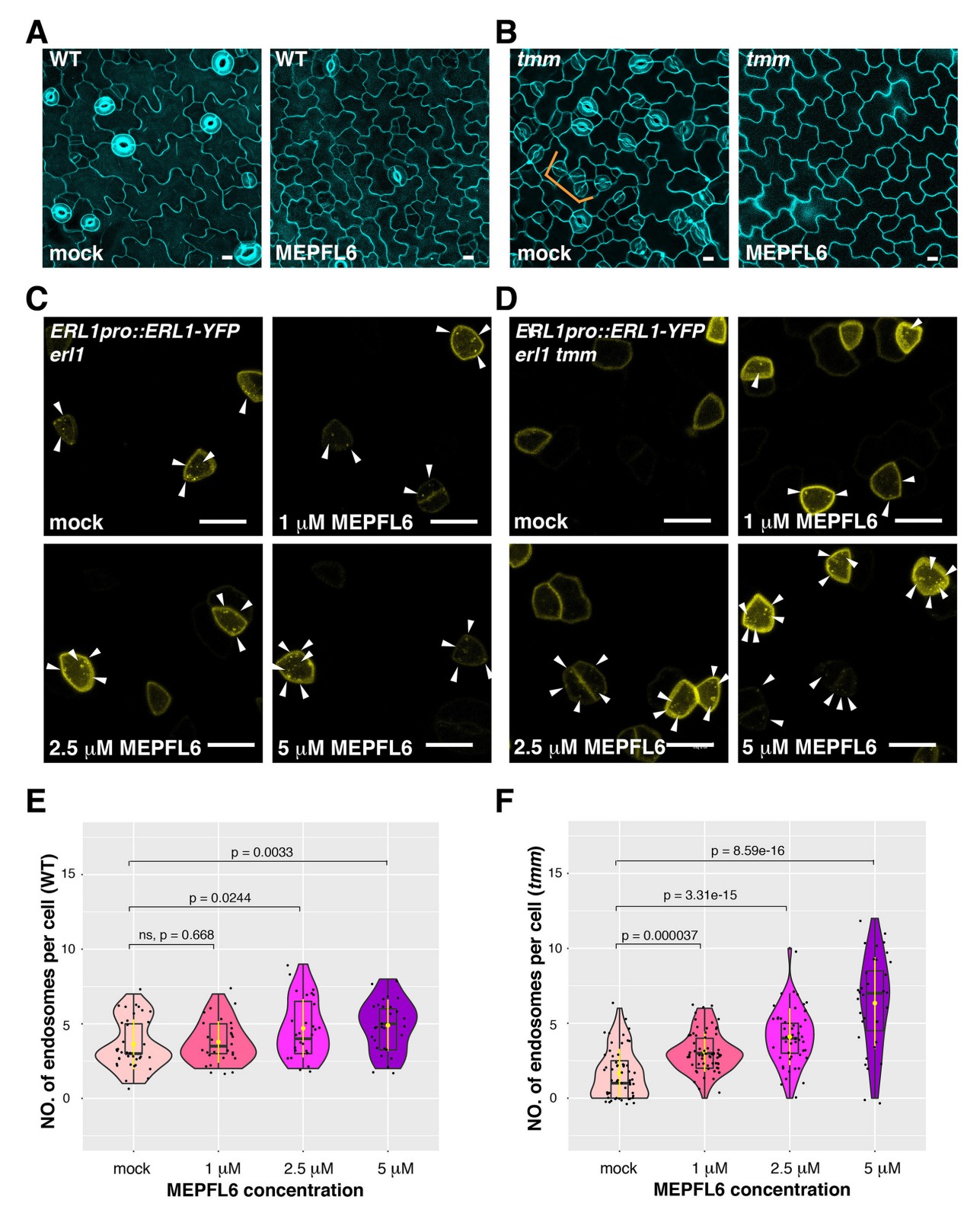

**Figure 6.** MEPFL6 triggers ERL1-YFP internalization in both *erl1* and *erl1 tmm*. (**A**) Representative confocal microscopy images of cotyledon abaxial epidermis from the 5-day-old wild type seedlings treated with mock (left) or 5 µM MEPFL6 (right). Scale bars = 10 µm. (**B**) Shown are representative confocal microscopy images of cotyledon abaxial epidermis from the 5-day-old *tmm* seedlings treated with mock (left) or 5 µM MEPFL6 (right). Scale bar = 10 µm. (**C**) Representative images of ERL1-YFP in *erl1* treated with mock (top left), 1 µM MEPFL6 (top right), 2.5 µM MEPFL6 (bottom left) and 5

*Figure 6 continued on next page*

Figure 6 continued

µM MEPFL6 (bottom right) are shown. Arrowheads indicate endosomes. Scale bar = 10 µm. (D) Representative images of ERL1-YFP in *erl1 tmm* treated with mock (top left), 1 µM MEPFL6 (top right), 2.5 µM MEPFL6 (bottom left) and 5 µM MEPFL6 (bottom right) are shown. Arrowheads indicate endosomes. Scale bars = 10 µm. (E) Quantitative analysis of the number of ERL1-YFP-positive endosomes per cell at different concentrations of MEPFL6 application in *erl1 are* shown as a Violin plot. Median values are shown as lines in the boxplot, and mean values are shown as yellow dots in the plot. Dots, individual data points. Welch's two sample T-test was performed for pairwise comparisons of samples treated with the mock and different concentration of MEPFL6. Number of cells analyzed, n = 37, 28, 27, 30 for treatment with mock, 1 µM, 2.5 µM, 5 µM MEPFL6. (F) Quantitative analysis of the number of ERL1-YFP-positive endosomes per cell at different concentrations of MEPFL6 application in *erl1 tmm* are shown as a Violin plot. Dots, individual data points. Median values are shown as lines in the boxplot, and mean values are shown as yellow dots in the plot. Welch's two sample T-test was performed for a pairwise comparisons of samples treated with the mock and different concentration of MEPFL6. Number of cells analyzed, n = 55, 63, 48, 35 for treatment with mock, 1 µM, 2.5 µM, 5 µM MEPFL6.

The online version of this article includes the following figure supplement(s) for figure 6:

**Figure supplement 1.** Stomatal development in *tmm* is more sensitive than in wild type to MEPFL6 application.

proteins from *erecta erl1 erl2* triple mutant seedlings rescued by *ERL1pro::ERL1-FLAG* were subjected to Endo-H treatment (see Methods). Under normal conditions, ERL1-FLAG is detected as a single band on immunoblots (*Figure 7D*, black arrow). The Endo-H digestion resulted in a faster mobility of ERL1-FLAG protein with at least three different sizes, suggestive of heterogeneous glycans (*Figure 7D*, blue and cyan arrows). In contrast, ERL1-FLAG protein from Stomagen-treated seedlings was hypersensitive to Endo-H and cleaved completely (*Figure 7D*, light arrow). Likewise, the *tmm* mutation enhanced the Endo-H sensitivity of ERL1 (*Figure 7—figure supplement 1B*), suggesting an increase in endoplasmic reticulum retention.

Because exogenous application of Stomagen blocks the activation of ERECTA-family signaling (*Lee et al., 2015*) and results in stomatal clustering (*Figure 7A*), we sought to address if insufficient internalization of ERL1 from the plasma membrane triggers its stalling in the endoplasmic reticulum. For this purpose, we first treated *erecta erl1 erl2* seedlings expressing ERL1-YFP with Tyr A23, an inhibitor that has been widely used to block clathrin-mediated endocytosis in plant cells (*Banbury et al., 2003*; *Santuari et al., 2011*). Indeed, the Tyr A23 treatment enhanced ERL1-YFP signals in the endoplasmic reticulum (*Figure 7E*, pink arrow), whereas in mock ERL1-YFP was only detected on the plasma membrane and endosomes.

A recent report showed that Tyr A23 functions as a protonophore, which inadvertently blocks endocytosis through cytoplasmic acidification (*Dejonghe et al., 2016*). Chemical screening and subsequent derivatization identified ES9-17 as a specific inhibitor of clathrin-mediated endocytosis without the side effects of cytoplasmic acidification (*Dejonghe et al., 2019*). We sought to test the effects of ES9-17 on ERL1-YFP subcellular localization to rule out the possibility that retention of ERL1-YFP in the endoplasmic reticulum is due to cellular acidification. ES9-17 previously has been applied only to root cells (*Dejonghe et al., 2019*). We first optimized the treatment condition for developing seedling shoots (see Methods). At 100 µM, ES9-17 inhibited the internalization of FM4-64 dye in epidermal pavement cells and stomatal-lineage cells, just like 50 µM ES9-17 in root cells (*Figure 7—figure supplement 3A,B*). Under this condition, ES9-17 treatment caused the accumulation of ERL1-YFP in the endoplasmic reticulum, just like the Tyr A23 treatment (*Figure 7E*). Taken together, our cell biological, pharmacological, and biochemical analyses reveal that inefficient endocytosis due to perception of an antagonistic peptide, Stomagen, as well as loss of co-receptor TMM, causes retention of ERL1-YFP in the endoplasmic reticulum.

## Cytoplasmic domain of ERL1 is required for proper subcellular trafficking behavior

Removal of the entire cytoplasmic domain from ERECTA-family RLKs is known to confer strong dominant-negative effects both in aboveground organ growth and in stomatal patterning (*Shpak et al., 2003*; *Lee et al., 2012*). The dominant-negative ERL1ΔK can directly bind its ligand EPF1 through the extracellular LRR domain. However, it is unable to signal and, consequently, confers paired and clustered stomata, thereby phenocopying *epf1* mutant (*Figure 8A,B*; *Lee et al., 2012*).

To gain further insight into the mechanism behind the complex subcellular dynamics of ERL1 upon different EPF/EPFL peptide perception, we examined the subcellular behaviors of the dominant-negative ERL1, ERL1ΔK-CFP driven by the endogenous *ERL1* promoter (*Figure 8*). Strong CFP

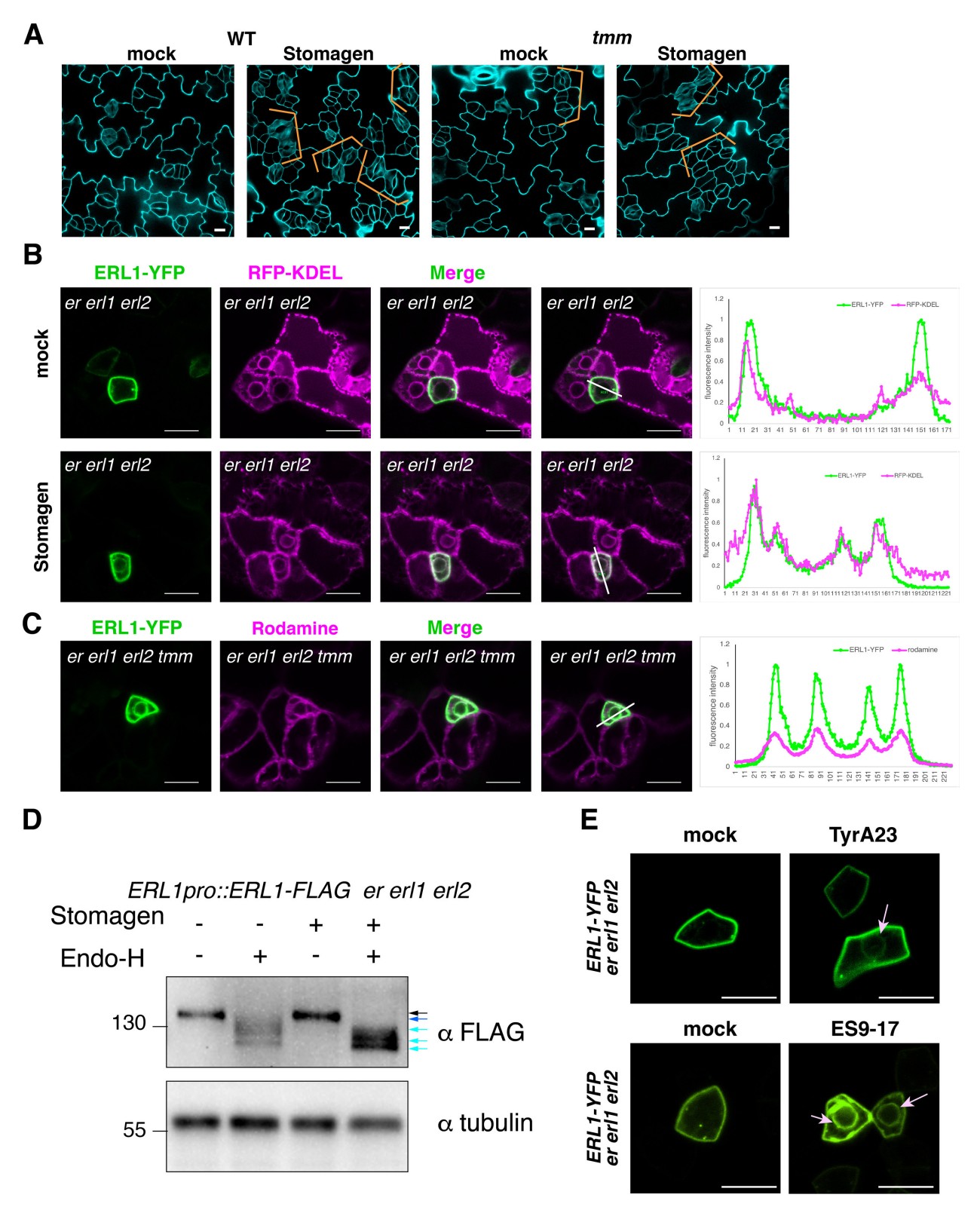

**Figure 7.** Stomagen application confers accumulation of ERL1 in endoplasmic reticulum. (**A**) Representative confocal microscopy images of cotyledon abaxial epidermis from the 5-day-old wild type seedlings (top) or *tmm* seedlings (bottom) treated with mock (left) or 5 µM Stomagen (right). Scale bars = 10 µm. (**B**) Representative confocal microscopy images of ERL1-YFP (left column) and co-localization analysis with the endoplasmic reticulum marker RFP-KDEL (middle column) in the abaxial epidermis of cotyledons of the 5-day-old *erecta* (*er*) *erl1 erl2* seedlings treated with mock (top row) or
*Figure 7 continued on next page*

Figure 7 continued

5 µM Stomagen (bottom row). Right, merged images with the line slicing along which quantification analysis of the YFP intensity (green) and RFP intensity (magenta) was done; graphs are shown on the right, with two middle peaks (pointed by arrowheads) showing signals from the endoplasmic reticulum and two big peaks on both sides showing signals of the plasma membrane. Scale bars = 10 µm. (C) Representative confocal microscopy images of ERL1-YFP (left) in the abaxial epidermis of cotyledons of the 5-day-old *erecta erl1 erl2* tmm seedlings stained with the endoplasmic reticulum dye Rhodamine (second left column). The merged image is shown in the third left column. Quantification analysis of the YFP intensity (green) and RFP intensity (magenta) along the line drawn in the right image is shown as a graph on the right, with two middle peaks (pointed by arrowheads) showing signals from the endoplasmic reticulum and two big peaks on both sides showing signals of the plasma membrane. Scale bars = 10 µm. (D) Immunoblot analysis of 3-day-old ERL1-FLAG *erecta erl1 erl2* seedlings treated with mock or 5 µM Stomagen for 2 days and then digested without or with Endo-H. Top panel shows the ERL1-FLAG detected by α-FLAG. Lower panel shows the loading control of Tubulin detected by α-Tubulin. Black arrow, ERL1 band without digestion, blue and cyan arrows, ERL1 cut with Endo-H digestion. (E) Representative confocal microscopy images of abaxial epidermis of true leaves from ERL1-YFP in *erecta erl1 erl2* seedlings treated with mock (top left) or 50 µM Tyr A23 (top right); mock (bottom left) or 100 µM ES9-17 (bottom right). Arrow indicates the ring-like structure, characteristics of endoplasmic reticulum localization, detected after treatment with Tyr A23 or ES9-17. Scale bars = 10 µm.

The online version of this article includes the following figure supplement(s) for figure 7:

**Figure supplement 1.** Inefficient endocytosis causes ERL1-YFP to stall in the endoplasmic reticulum.
**Figure supplement 2.** Stomagen application does not affect the subcellular localization of BRI1-GFP.
**Figure supplement 3.** ES9-17 inhibits endocytosis of leaf epidermal cells.

signal is detected on the plasma membrane of stomatal precursor cells, but only very few mobile punctae can be seen in the cytoplasm (*Figure 8C*). BFA treatment results in 86% cells possessing an average of 2 ERL1ΔK-CFP-labelled BFA bodies (*Figure 8C*, *Figure 8—figure supplement 1A*). However, the BFA sensitivity of ERL1ΔK was also observed in the presence of CHX in a statistically-indistinguishable manner from those in *tmm* mutant (*Figure 4—figure supplement 1*, p=0.412), supporting that the dominant-negative form of ERL1 may confer reduced endocytosis/internalization. Moreover, ERL1ΔK-CFP exhibits insensitivity to Wm treatment, with only 18% cells showing few Wm bodies highlighted by ERL1ΔK-CFP (*Figure 8—figure supplement 1A*). The reduced endocytosis of ERL1ΔK-CFP is not due to defects in the general endocytosis process, as FM4-64 can still internalize in the ERL1ΔK-CFP-positive cells on the transgenic seedling epidermis, like it does in cells with the full-length ERL1 (*Figure 8—figure supplement 1B*).

Next, we examined the subcellular behaviors of ERL1ΔK-CFP upon perception of EPF1 and Stomagen peptides. As shown in *Figure 8D and E*, application of 5 µM MEPF1 did not increase endocytosis of ERL1ΔK-CFP (p=0.9972), whereas the same treatment triggered the endocytosis of control ERL1-YFP (p=0.0002) (*Figure 8D,E*). Thus, the dominant-negative ERL1 is insensitive to EPF1 peptide ligand. Together with the case of ERL1-YFP in *tmm* mutant (*Figure 5*), the results suggest that the internalization of ERL1 by EPF1 peptide reflects the active signaling events.

To address whether ERL1ΔK-CFP behaves similarly to signaling-incompetent full-length ERL1-YFP (i.e. in *tmm erecta erl1 erl2* background), we next applied Stomagen peptide. To our surprise, unlike ERL1-YFP (*Figure 8F*), ERL1ΔK-CFP failed to accumulate in the endoplasmic reticulum and remained insensitive to Stomagen peptide (*Figure 8G*). Combined, these results indicate that the dominant-negative ERL1ΔK is dysfunctional in ligand-induced subcellular trafficking and suggest that the cytoplasmic domain of ERL1 is critical for its proper subcellular dynamics.

## Discussion

In this study, we revealed that ERL1 endocytosis accurately reflects EPF signal perception based on three pieces of evidence (*Figure 9*): first, both EPF1 and EPFL6 peptides trigger ERL1 endocytosis. Second, in the absence of the co-receptor TMM, ERL1 endocytosis is compromised and becomes insensitive to EPF1 application. Third, the cytoplasmic domain of ERL1 is required for ERL1 endocytosis. EPF1 and EPFL6 peptide application increased ERL1 population in endosomes in a dosage-dependent manner (*Figures 5* and *6*). Our pharmacological application of Wortmannin, BFA, BFA with CHX and Concanamycin A (*Figures 2–3*, *Figure 4—figure supplement 1*), indicate that ERL1 is constitutively recycled whereas the receptor activation triggers endocytosis for vacuolar sorting via MVB/LE. In this aspect, the subcellular dynamics of ERL1 resembles that of FLS2, which is also constitutively recycled but rapidly removed from the cell surface upon flg22 perception (*Robatzek et al.*,

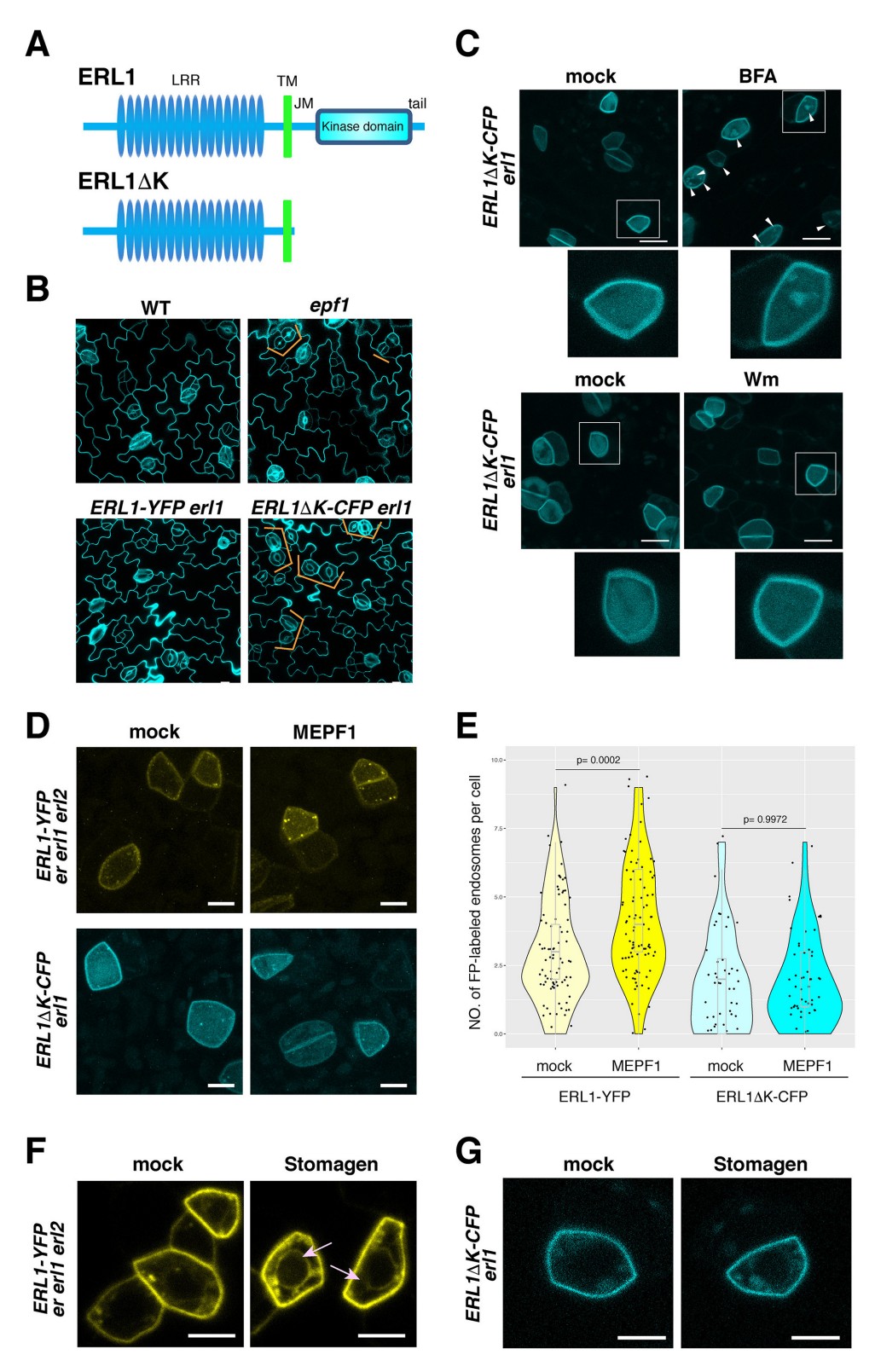

**Figure 8.** Dominant-negative form of ERL1 is compromised in subcellular trafficking. (**A**) Diagram of the full-length ERL1 protein (top) and the dominant-negative ERL1 protein lacking the cytoplasmic domain (bottom). (**B**) Representative confocal microscopy images of cotyledon abaxial epidermis from 4-day-old seedlings of wild type, *epf1*, ERL1-YFP *erl1* and ERL1ΔK-CFP *erl1*, stained by PI. Orange brackets indicate the paired stomata in *epf1* and ERL1ΔK -CFP in *erl1*. Scale bars = 10 µm. (**C**) Representative confocal microscopy images of ERL1ΔK-CFP treated with mock (top left for

*Figure 8 continued on next page*

Figure 8 continued

BFA treatment), 30 µM BFA (top right), mock (bottom left for Wm treatment) and 25 µM Wm (bottom right). Inset, enlarged image of a representative meristemoid. Arrowheads indicate BFA bodies. Scale bars = 10 µm. (D) Representative confocal microscopy images of an abaxial true leaf epidermis from seedlings expressing ERL1-YFP in *erecta erl1 erl2* (top) and ERL1ΔK-CFP in *erl1* (bottom) treated with mock (left) or 5 µM MEPF1. Scale bars = 5 µm. (E) Quantitative analysis of the number of ERL1-YFP-positive or ERL1ΔK-CFP-positive endosomes per cell in mock or upon 5 µM MEPF1 peptide application. Dots, individual data points with a jitter (0.2). A box plot is overlaid to each violin plot, with a median shown as a line. T-test was performed for pairwise comparisons of samples treated with the mock and MEPF1. Two-way ANOVA analysis: genotype (ERL1-YFP, ERL1ΔK-CFP), p=7.081e-12; treatment (mock, MEPF1), p=0.001498. Tukey's HSD, genotype: treatment, ERL1-YFP mock: ERL1-YFP MEPF1 treatment, p=0.00045, ERL1ΔK-CFP mock: ERL1ΔK-CFP EPF1 treatment, p=0.9998. Number of cells analyzed, n = 88 (ERL1-YFP, mock), 92 (ERL1-YFP, MEPF1), 50 (ERL1ΔK-CFP, mock), 59 (ERL1ΔK-CFP, MEPF1). (F) Representative confocal microscopy images of abaxial epidermis of true leaves from ERL1-YFP expressed in *erecta erl1 erl2* seedlings treated with mock or 5 µM Stomagen peptide.). Arrow indicates the ring-like structure, characteristics of endoplasmic reticulum localization. Scale bars = 5 µm. (G) Representative confocal microscopy images of abaxial epidermis of true leaves from ERL1ΔK-CFP in *erl1* seedlings treated with mock or 5 µM Stomagen peptide. Scale bars = 5 µm.

The online version of this article includes the following figure supplement(s) for figure 8:

**Figure supplement 1.** Dominant-negative ERL1ΔK-CFP is predominantly localized at plasma membrane.

*2006*; *Smith et al., 2014*). Unlike FLS2, however, a vast majority of ERL1-YFP signal remained at the plasma membrane even after treatment of 5 µM MEPF1 (*Figure 5*). These differences could be attributed to the roles of FLS2 and ERL1 in immunity *vs.* development, respectively. FLS2 mediates acute pathogen-induced defense response, whereas ERL1 likely detects endogenous peptides to influence slower processes of cell division and differentiation. A recent study showed, however, that defects in the clathrin-mediated FLS2 endocytosis impair only a subset of FLS2-mediated immune responses (*Mbengue et al., 2016*). Thus, the precise contributions of endocytosis and cellular response remain open questions.

It has been shown that EPF1, but not EPFL6, requires TMM for the inhibition of stomatal development (*Hara et al., 2007*; *Abrash and Bergmann, 2010*). Likewise, structural analyses of the EPF-ERECTA family complexes showed that EPF1, but not EPFL6, requires TMM for binding to the ectodomain of ERECTA family receptors (*Lin et al., 2017*). Here, we demonstrate that TMM is required for endocytosis triggered by EPF1, but not by EPFL6 (*Figures 4*, *5*, *6* and *9*). Thus, at least two populations of ERL1 receptor complexes must be present on the plasma membrane, with and without TMM. The BFA treatment on *tmm* mutants (*Figure 4*, *Figure 4—figure supplement 1*) revealed that in the absence of TMM, ERL1-YFP internalization is compromised but not absent. This is most likely due to the presence of EPF-LIKE peptide (e.g. EPFL4/5/6) family members that do not require TMM. The FRAP analysis indeed detected the different mobility of these two ERL1 compositions on the plasma membrane (*Figure 4*). Multiple compositions of receptor complexes have also been reported in CLV3 signaling, where CLV1 homomers, CLV2/CORYNE (CRN) heterodimers and CLV1/CLV2/CRN multimers co-exist on the plasma membrane (*Somssich et al., 2015*). However, only the microdomain-localized CLV1/CLV2/CRN multimers can perceive the sole ligand CLV3. In the case of BRI1 and FLS2, pre-formed BRI1-BAK1 complex was detected regardless of BRs whereas FLS2 forms FLS2-BAK1 complex upon flg22 application (*Bücherl et al., 2013*; *Somssich et al., 2015*). These receptor complexes are spatially separated, even though BRI1 and FLS2 share the same co-receptor BAK1 (*Bücherl et al., 2013*; *Somssich et al., 2015*; *Bücherl et al., 2017*; *Hutten et al., 2017*). On the contrary, both populations of ERL1 complexes are 'functional' and ligand-inducible, as they can perceive EPF1 or EPFL6, respectively (*Figures 5* and *6*). It is possible that the distinct ERL1 receptor complexes reside in different microdomains on the plasma membrane and undergo different trafficking routes upon the correlated ligand perception. EPF1 triggers ERL1 association with BAK1 (*Meng et al., 2015*). Examining spatiotemporal subcellular dynamics of ERL1 together with TMM and BAK1 at a super resolution scale may reveal the contribution of each receptor complex for specific signal perception and transduction.

ERL1 is retained in the endoplasmic reticulum when treated with exogenous Stomagen. This ERL1 subcellular behavior is associated with its own biological contexts, given that BRI1, an LRR-RK with unrelated function, does not change its plasma membrane localization upon Stomagen application (*Figure 7—figure supplement 2*). Extensive studies support that the steady state of a protein in its subcellular compartment is interdependent on the anterograde and retrograde trafficking routes (*Brandizzi and Barlowe, 2013*). For example, a secretory protein is often retained in the

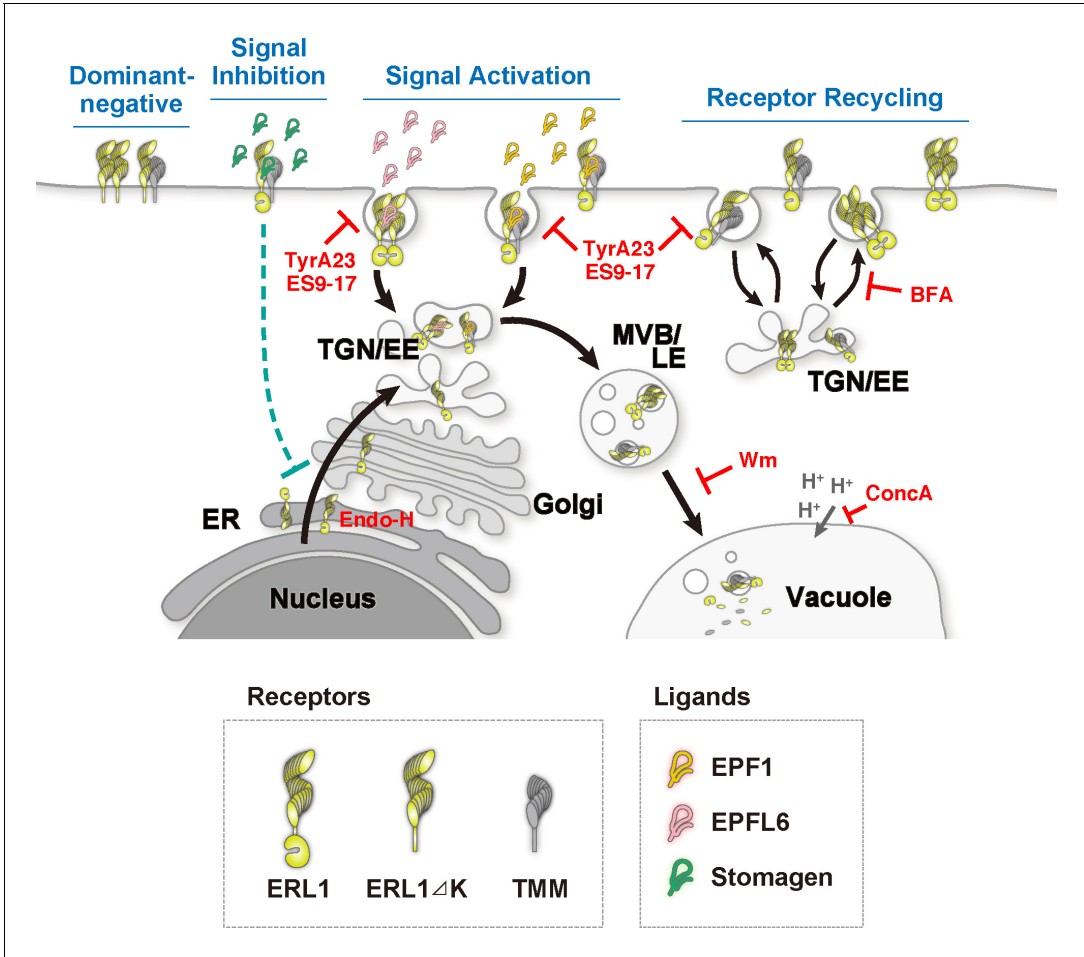

**Figure 9.** Schematic model of ERL1 subcellular dynamics triggered by diverse EPF peptides with different biological activities. ERL1 (light green) is constitutively recycling and follows BFA-sensitive endosomal pathway (Receptor Recycling). EPF1 (orange) and EPFL6 (pink) peptide ligands both activate ERL1 to inhibit stomatal differentiation, trigger ERL1 trafficking via Wm-sensitive MVB/LE to the vacuole (Signal Activation). EPF1-triggered ERL1 trafficking requires the presence of TMM (gray). In contrast, EPFL6 triggers ERL1 trafficking in TMM-independent manner. Stomagen (dark green), which blocks ERL1 signaling, causes stalling of ERL1 in endoplasmic reticulum (ER) (Signal Inhibition). The dominant-negative ERL1ΔK is overwhelmingly plasma-membrane localized and does not undergo normal subcellular trafficking (Dominant Negative). The site of action of each inhibitor/enzyme is indicated in red.

endoplasmic reticulum when the downstream secretion pathway is compromised (*Zheng et al., 2005*). Blocking the endoplasmic reticulum-to-Golgi retrograde trafficking will accelerate protein transport to the cell surface (*Fossati et al., 2014*). It is thus possible that Stomagen binding prevents the ERL1 endocytosis and the plasma membrane-accumulated ERL1 interferes with the normal transport of incoming ERL1 from the endoplasmic reticulum. Two additional pieces of evidence support this hypothesis. First, when endocytosis is blocked by Tyr A23 (*Banbury et al., 2003*) or ES7-19, the improved, specific inhibitor of clathrin heavy chain (*Dejonghe et al., 2019*), strong ERL1 signals become evident in the endoplasmic reticulum (*Figure 7E*). Second, in leaves of the *tmm* mutant, EPF1-triggered ERL1 endocytosis is compromised, ERL1 also accumulates in the endoplasmic reticulum (*Figure 7C* and *Figure 7—figure supplement 1*).

Stomagen-triggered ERL1 accumulation in endoplasmic reticulum may be highlighting the role of the endoplasmic reticulum-plasma membrane contact sites as a direct communication link between the two compartments (*Carrasco and Meyer, 2011*). The VAP-RELATED SUPPRESSOR OF TMM (VST) family plasma membrane proteins that interact with integral endoplasmic reticulum proteins, have been reported to facilitate ERECTA family-mediated signaling in stomatal development (*Ho et al., 2016*). Hence, Stomagen perception by the ERL1-TMM complex on the plasma

membrane may directly influence signaling via the contact sites and therefore affect the transport of ERL1 to the cell surface. The endoplasmic reticulum accumulation of ERL1-YFP is notable in the *erecta erl1 erl2* mutant background, that is when functional *ERECTA* and *ERL2* genes are missing. Because Stomagen interacts with ERECTA at the early stage of stomatal initiation (*Lee et al., 2015*), the absence of *ERECTA* likely confers sensitized, exaggerated response of ERL1-YFP to Stomagen. Alternatively, a dysregulated cell-cell signaling in the *erecta erl1 erl2* background may abrogate the buffering of exogenously applied Stomagen peptide.

Posttranslational modifications, such as phosphorylation and ubiquitination, of the receptors have emerged as key regulatory mechanisms of receptor subcellular dynamics in FLS2 and BRI1 (*Robatzek et al., 2006*; *Lu et al., 2011*; *Martins et al., 2015*; *Zhou et al., 2018*). Specific posttranslational modifications of ERL1 are yet unknown, but perception of different EPF/EPFL peptides may trigger unique phosphorylation codes that influence subsequent trafficking of ERL1. The dominant-negative ERL1 fails to respond to either MEPF1 or Stomagen peptides (*Figure 8*, *Figure 8—figure supplement 1*). Clearly, the behaviors of ERL1ΔK-CFP is different from the full-length ERL1-YFP. The accumulation of ERL1 in the endoplasmic reticulum occurs when ERL1-YFP becomes signaling incompetent- (i) by Stomagen peptide perception; (ii) due to the absence of co-receptor and redundant receptors, TMM, ERECTA, and ERL2; or (iii) when ERL1 endocytosis is pharmacologically blocked (*Figure 7*). Thus, subcellular dynamics of the full-length ERL1 is highly tuned and coordinated. Such coordination is lost by removal of the cytoplasmic domain, which includes juxtamembrane domain, kinase domain, and C-terminal tail region (*Figure 8A*). It is highly plausible that ERL1ΔK-CFP can no longer be recognized as the original endocytosis cargo by adaptor proteins. For instance, the endocytosis of BRI1 is mediated by direct association with Clathrin adaptor complex AP2 (*Di Rubbo et al., 2013*). Future studies may reveal the molecular basis of how ERL1 interacts with specific cargo receptors.

Previous work reported that BFA effect (BFA body formation) is different in root vs. shoot tissues (*Robinson et al., 2008*; *Langhans et al., 2011*). Here, we extensively documented the effects of BFA on subcellular behaviors using eight endomembrane organelle markers together with endocytic tracer dye (*Figure 3*, *Figure 3—figure supplement 1*). Our extensive analyses show that with appropriate concentration and treatment condition, BFA body formation can be recapitulated in leaf cells, just like in root cells. The results could facilitate future studies on recycling and endocytosis of membrane proteins in intact leaves. Taken together, our work revealed the mechanism by which multiple peptide ligands with distinct activities, EPF1, EPFL6, and Stomagen, fine-tune stomatal patterning at the level of the subcellular dynamics of a single receptor, ERL1. Successful development of visible functional peptide ligands and identification of the immediate biochemical events by the ERECTA-family perceiving different EPF peptides will help elucidate the exact roles of receptor trafficking and signaling specifying developmental patterning in plants.

## Materials and methods

### Key resources table

| Reagent type (species) or resource | Designation | Source or reference | Identifiers | Additional information |
|---|---|---|---|---|
| Genetic reagent (*Arabidopsis thaliana*) | *erecta* (*er-105*) | *Shpak et al., 2005* PMID:16002616 | | See Materials and method, section 1 |
| Genetic reagent (*Arabidopsis thaliana*) | *erl1-2* | *Shpak et al., 2005* PMID:16002616 | | See Materials and method, section 1 |
| Genetic reagent (*Arabidopsis thaliana*) | *erl2-1* | *Shpak et al., 2005* PMID:16002616 | | See Materials and method, section 1 |
| Genetic reagent (*Arabidopsis thaliana*) | *epf1-1* | *Hara et al., 2007* PMID:17639078 | | See Materials and method, section 1 |
| Genetic reagent (*Arabidopsis thaliana*) | *tmm-KO* | *Hara et al., 2007* PMID:17639078 | | See Materials and method, section 1 |
| Genetic reagent (*Arabidopsis thaliana*) | *ERL1pro::ERL1-YFP in erl1-2* | *Lee et al., 2012* PMID:22241782 | | See Materials and method, section 1 |

*Continued on next page*

*Continued*

| Reagent type (species) or resource | Designation | Source or reference | Identifiers | Additional information |
|---|---|---|---|---|
| Genetic reagent (*Arabidopsis thaliana*) | *ERL1pro::ERL1-YFP in erl1-2 tmm* | This study | | See Materials and method, section 1 |
| Genetic reagent (*Arabidopsis thaliana*) | *ERL1pro::ERL1-YFP in erl1-2 epf1* | This study | | See Materials and method, section 1 |
| Genetic reagent (*Arabidopsis thaliana*) | *ERL1pro::ERL1-FLAG in erl1-2* | *Lee et al., 2012* PMID:22241782 | | See Materials and method, section 1 |
| Genetic reagent (*Arabidopsis thaliana*) | *ERL1pro::ERL1-FLAG in erecta erl1-2 erl2-1* | *Lee et al., 2012* PMID:22241782 | | See Materials and method, section 1 |
| Genetic reagent (*Arabidopsis thaliana*) | *ERL1pro::ERL1ΔK-CFP in erl1-2* | *Lee et al., 2012* PMID:22241782 | | See Materials and method, section 1 |
| Genetic reagent (*Arabidopsis thaliana*) | *MUTEpro::ERL1-YFP in er-105 erl1-2 erl2-1* | *Qi et al., 2017* PMID:28266915 | | See Materials and method, section 1 |
| Genetic reagent (*Arabidopsis thaliana*) | *iEPF1* | *Qi et al., 2017* PMID:28266915 | | See Materials and method, section 1 |
| Genetic reagent (*Arabidopsis thaliana*) | *BRI1pro::BRI1-GFP* | *Friedrichsen et al., 2000* PMID:10938344 | | See Materials and method, section 1 |
| Genetic reagent (*Arabidopsis thaliana*) | *UBQ10pro::YFP-SYP32* | *Geldner et al., 2009* PMID:19309456 | Wave22Y | See Materials and method, section 1 |
| Genetic reagent (*Arabidopsis thaliana*) | *UBQ10pro::YFP-ARA7* | *Geldner et al., 2009* PMID:19309456 | Wave2Y | See Materials and method, section 1 |
| Genetic reagent (*Arabidopsis thaliana*) | *UBQ10pro::YFP-RabA5d* | *Geldner et al., 2009* PMID:19309456 | Wave24Y | See Materials and method, section 1 |
| Genetic reagent (*Arabidopsis thaliana*) | *UBQ10pro::YFP-RabA1e* | *Geldner et al., 2009* PMID:19309456 | Wave34Y | See Materials and method, section 1 |
| Genetic reagent (*Arabidopsis thaliana*) | *SYP43pro::GFP-SYP43* | *Uemura et al., 2012* PMID:22307646 | | See Materials and method, section 1 |
| Genetic reagent (*Arabidopsis thaliana*) | *SYP22pro:: SYP22- mRFP* | *Ebine et al., 2011*; *Postma et al., 2016*, PMID:21666683, 26765243 | | See Materials and method, section 1 |
| Genetic reagent (*Arabidopsis thaliana*) | *ARA7pro::mRFP-ARA7* | *Ebine et al., 2011*; *Postma et al., 2016*, PMID:21666683, 26765243 | | See Materials and method, section 1 |
| Genetic reagent (*Arabidopsis thaliana*) | *SYP43pro:: SYP43- mRFP* | *Ebine et al., 2011*; *Postma et al., 2016*, PMID:21666683, 26765243 | | See Materials and method, section 1 |
| Genetic reagent (*Arabidopsis thaliana*) | *ARA6pro::ARA6-GFP* | *Ebine et al., 2011* PMID:21666683 | | See Materials and method, section 1 |
| Genetic reagent (*Arabidopsis thaliana*) | *ST-RFP* | *Faraco et al., 2017* PMID:28636930 | | See Materials and method, section 1 |
| Genetic reagent (*Arabidopsis thaliana*) | *KDEL-RFP* | *Faraco et al., 2017* PMID:28636930 | | See Materials and method, section 1 |
| Genetic reagent (*Arabidopsis thaliana*) | *CaMV35S::N-ST-YFP* | *Takagi et al., 2013* PMID:24280388 | | See Materials and method, section 1 |
| Genetic reagent (*Arabidopsis thaliana*) | *CaMV35S::GFP-HDEL* | *Mitsuhashi et al., 2000* PMID:11100771 | | See Materials and method, section 1 |
| Chemical compound, drug | Brefeldin A (BFA) | Cayman Chemical | Cat. #: 11861 Lot #: 0533646–12 | See Materials and method, section 3 |
| Chemical compound, drug | Concanamycin A | Adipogen Life Sceinces | Cat. #: BVT-0237-C025 Lot #: B1105151 | See Materials and method, section 3 |
| Chemical compound, drug | Brefeldin A (BFA) | Sigma | Cat. #: B7651 | See Materials and method, section 3 |

*Continued on next page*

*Continued*

| Reagent type (species) or resource | Designation | Source or reference | Identifiers | Additional information |
|---|---|---|---|---|
| Chemical compound, drug | Wortmannin (Wm) | Sigma | Cat. #: W1628 | See Materials and method, section 3 |
| Chemical compound, drug | Cycloheximide (CHX) | Sigma | Cat. #: C4859 | See Materials and method, section 3 |
| Chemical compound, drug | Rhodamine B | Sigma | Cat. #: R6626 | See Materials and method, section 3 |
| Chemical compound, drug | Tyrphostin (Tyr A23) | Sigma | Cat. #: T7165 | See Materials and method, section 3 |
| Chemical compound, drug | ES9-17 | *Dejonghe et al., 2019* PMID:31011214 | | See Materials and method, section 3 |
| Peptide, recombinant protein | MEPF1 | *Lee et al., 2012* PMID:22241782 | pJSL11 | See Materials and method, section 2 |
| Peptide, recombinant protein | MEPFL6 | This work | pJSL79 | See Materials and method, section 2 |
| Peptide, recombinant protein | STOMAGEN | *Lee et al., 2015* PMID:26083750 | | See Materials and method, Section 2 |
| Antibody | Mouse monoclonal anti-FLAG M2 | Sigma | Cat. #: F-3165 Lot #: 065K6236 | WB (1:5000 dilution) |
| Antibody | Horseradish peroxidase-conjugated goat anti-mouse IgG | GE Healthcare | Cat. #: NA931VS Lot #: 9708060 | WB (1:50,000 dilution) |
| Antibody | Mouse anti-tubulin | Millipore Sigma | Cat. #: MABT205 Lot #: 2999783 | WB (1:5000 dilution) |
| Software, algorithm | R (ver 3.4.1) | *Ritz et al., 2015* PMID:26717316 | | See Materials and method, section 2 and section 7 |
| Software, algorithm | Leica LAS AF | http://www.leica-microsystems.com | | See Materials and method, section 5 |
| Software, algorithm | Fiji | https://imagej.net/Fiji | | See Materials and method, section 5 |
| Software, algorithm | Imaris 8.1 | Bitplane | | See Materials and method, section 5 |
| Software, algorithm | FrapBot | *Kohze et al., 2017* www.frapbot.kohze.com | | See Materials and method, section 6 |
| Other | FM4-64 | Invitrogen/Thermo Fisher Scientific | Cat. #: T13320 | See Materials and method, section 3 |
| Other | Endo-H | NEB | Cat. #: P0703S | See Materials and method, section 4 |

## Plant materials and growth conditions

The *Arabidopsis* accession Columbia (Col) was used as wild type. The following mutants and reporter transgenic plant lines were reported previously: *erecta* (*er-105*) (*Shpak et al., 2005*); *erl1-2* (*Shpak et al., 2005*); *erl2-1* (*Shpak et al., 2005*); *epf1-1* (*Hara et al., 2007*); *tmm-KO* (*Hara et al., 2007*); *ERL1pro::ERL1-YFP* in *erl1-2*, *ERL1pro:: ERL1-FLAG in erl1-2* and *erecta erl1-2 erl2-1*, and *ERL1pro::ERL1ΔKinase* in *erl1-2* (*Lee et al., 2012*); *MUTEpro::ERL1-YFP* in *er-105 erl1-2 erl2-1* and *iEPF1* lines (*Qi et al., 2017*); *BRI1pro::BRI1-GFP* (*Friedrichsen et al., 2000*), *UBQ10pro::YFP-SYP32* (Wave22Y), *UBQ10pro::YFP-RabA5d* (Wave24Y), *UBQ10pro::YFP-RabA1e* (Wave34Y), and *UBQ10-pro::YFP-ARA7* (Wave2Y) (*Geldner et al., 2009*); *SYP43pro::GFP-SYP43* (*Uemura et al., 2012*), *ARA6pro::ARA6-GFP* (*Ebine et al., 2011*), *ARA7pro::mRFP-ARA7*, *SYP22pro::mRFP-SYP22*, and *SYP43pro::mRFP-SYP43* (*Ebine et al., 2011*; *Postma et al., 2016*) are a gift from Prof. Takashi Ueda (NIBB, Japan); *ST-RFP* and *KDEL-RFP* constructs (*Faraco et al., 2017*) are from Prof. Gian Pietro Di Sansebastiano (Univ. of Salento, Italy); *35Spro::GFP-HDEL* (*Mitsuhashi et al., 2000*) and *CavM35S::*

ST-YFP (*Takagi et al., 2013*) are from Prof. Ikuko Hara-Nishimura (Konan University, Japan); Reporter lines were introduced into respective mutant backgrounds by genetic crosses or by *Agrobacterium*-mediated floral-dipping transformation, and genotypes were confirmed by PCR. Seedlings and plants were grown as described previously (*Lee et al., 2012*). For a list of PCR-based genotyping primer sequence, see Table S1.

## Recombinant peptide production

For recombinant peptide production of EPFL6, a coding sequence covering a predicted mature EPFL6 (MEPFL6) peptide was cloned into pBADgIII vector to make a fusion construct with 6xHis-tag (pJSL79: generated by Dr. Jin Suk Lee). Expression, purification, and refolding of predicted mature EPF1 (MEPF1) or EPFL6 (MEPFL6) peptides were performed as described previously (*Lee et al., 2012*), except for the following. His-tagged MEPF1 or MEPFL6 was affinity purified on 5 ml His-Trap HP column (GE Healthcare) using NGC Chromatography System (Bio-Rad). Inclusion bodies from 1.0 L of *E. coli* were solubilized in guanidine hydrochloride (Gdn-HCl) buffer (6.0 M Gdn-HCl, 500 mM NaCl, 5 mM imidazole, 1 mM 2-mercaptoethanol, 50 mM Tris, pH 8.0) and loaded onto the column and washed with 10 column volumes (50 mL) of Wash Buffer (8.0 M urea, 500 mM NaCl, 30 mM imidazole, 1 mM β-mercaptoethanol, 50 mM Tris, pH 8.0) at a flow rate of 3.00 ml/min, and MEPF1 or MEPFL6 peptides were eluted with a 0–100% gradient of Wash to Elution Buffer (8.0 M urea, 500 mM NaCl, 500 mM imidazole, 1 mM β-mercaptoethanol, 50 mM Tris, pH 8.0) over 10 column volumes at 3.00 mL/min prior to refolding. The quality of refolded peptide was analyzed by HPLC (Walters DataPrep 300), its bioactivity was confirmed using Arabidopsis seedlings, and bioassay on Arabidopsis seedlings were performed as described previously (*Lee et al., 2012*; *Lee et al., 2015*). Peptide treatments were performed as described previously (*Lee et al., 2015*). For a control experiment of Stomagen treatment on ERL1ΔK-CFP, ERL1-YFP driven by *MUTE* promoter in *erecta erl1 erl2* (*Qi et al., 2017*) was used due to the germination issues of ERL1-YFP driven by its own promoter. For the dose-response analysis of EPFL6, the R-package 'drc' (*Ritz et al., 2015*) was used to fit the binding curve to the generalized log logistic distribution (*Uchida et al., 2018*).

## Pharmacological treatment

Surface-sterilized seeds were sown on half-strength MS media containing 1% (w/v) sucrose and 1% (w/v) agar for 7 days in growth chamber. To synchronize germination, plates were placed in the dark at 4°C for 2 days. Plants were grown vertically in a 16 hr/8 hr light/dark cycle (40 µE m$^{-2}$ s$^{-1}$). Cotyledons were removed by scissors before drug treatment. For Concanamycin A treatment, plants were soaked in Milli-Q ultrapure-water (Merck Millipore) containing 2 µM FM4-64 (Thermo Fisher Scientific) with vacuum and then incubated in ultrapure water without FM4-64 for 6 hr to washout excessive dye. Then plants were transferred into ultrapure water containing 1 µM concanamycin A (AdipoGen Life Sciences, CA, USA) for 5 hr.

BFA (Sigma: Cat No. B7651) and Wortmannin (Sigma: Cat No. W1628) were dissolved as 10 mM stock using ethanol and DMSO, respectively. For BFA treatment, cotyledons of 7-day-old seedlings were removed, and the rest of the seedlings were immersed into either mock (0.3% of ethanol), or 30 µM BFA solution, vacuumed for 1 min, and immersed for 30 min before imaging. For a series of BFA concentration treatment experiments analyze its effect on endomembrane systems, whole plants were soaked in ultrapure water containing 5 µM FM4-64 and 30, 90, or 180 µM BFA (Cayman Chemical, MI, USA; Stock solution: 50 mM in DMSO) or equivalent concentration of solvent DMSO with vacuum.

For Wortmannin treatment, seedlings were treated with 25 µM Wortmannin in 0.25% DMSO. 0.25% DMSO solution was used as a mock condition. For MEPF1 and MEPFL6 treatment, purified peptide solution was diluted to 5 µM using liquid ½ MS media. Cotyledons of 7-day-old seedlings were removed, and the rest of the seedlings were immersed into the above solutions, vacuumed for 1 min, and immersed for 10 min before imaging. The same procedure was done for Stomagen treatment except that the seedlings were immersed into the solution for 1 hr. For co-treatment of cycloheximide (CHX: Sigma, C4859) and BFA, 7-day-old seedlings, with cotyledons moved, were immersed into 50 µM CHX for 1 hr followed by either mock (0.3% of ethanol), or 30 µM BFA solution, vacuumed for 1 min, and immersed for 30 min before imaging.

For Tyr A23 (Sigma: Cat No. T7165) treatment, Tyr A23 was dissolved as 50 mM stock using DMSO. 5-day-old seedlings were immersed into either mock (0.1% of DMSO) or 50 µM Tyr A23 solution, vacuumed for 1 min, and immersed for 1 hr before imaging.

ES9-17 was generously provided by Dr. Eugenia Russinova (VIB, Gent), and ES9-17 treatment was done as described in *Dejonghe et al., 2019*. Briefly, ES9-17 was dissolved as 50 mM stock using DMSO. For ES9-17 and FM 4–64 treatment on true leaves, cotyledons of 7-day-old seedlings were removed, and the rest of the seedlings were immersed into either mock (1/2 MS medium with 0.4% of DMSO) or 100 µM ES9-17 solution (1/2 MS medium with 50 µM ES9-17) for 30 min followed by 5 µM FM 4–64 (Thermo Fisher, T13320) staining for 30 min before imaging. For ES9-17 and FM 4–64 treatment in roots, 3-day-old seedlings were immersed into either mock (1/2 MS medium with 0.4% of DMSO), or 50 µM ES9-17 solution (1/2 MS medium with 100 µM ES9-17) for 30 min, followed by FM 4–64 (5 µM) staining for 30 min before imaging.

For Rhodamine B (Sigma: Cat No. R6626) hexyl ester treatment, Rhodamine B hexyl ester was dissolved as 16 mM stock using DMSO. 5-day-old seedlings were immersed into either mock (1% of DMSO) or 160 µM Rhodamine B hexyl ester solution for 30 min before imaging.

## Protein extraction, enzymatic assay (Endo-H), and protein gel immunoblot analysis

For Endo-H (NEB: Cat No. P0703S) assays, *erecta erl1 erl2* seedlings with functional *ERL1pro::ERL1-FLAG* were grown on ½ MS media plates for 3 days and then transferred to ½ MS liquid media with either Tris-HCl buffer (pH 8.8) or 5 µM Stomagen peptide in a 24-well cluster plate at room temperature for one day before being pooled for harvest. Plant materials were ground in liquid nitrogen, and then extracted with buffer (100 mM Tris-HCl pH 8.8, 150 mM NaCl, 1 mM EDTA, 20% glycerol, 20 mM NaF, 2 mM $Na_3VO_4$, 1 mM PMSF, 1% Triton X-100, 1 tablet per 50 ml extraction buffer of cOmplete proteinase inhibitor cocktail, Roche). The extracts were briefly sonicated at 4°C and centrifuged at 4000 r.p.m. for 10 min at 4°C to remove cell debris. The supernatant was then ultracentrifuged at 100,000 g for 30 min at 4°C. Total protein concentration was determined using a Bradford assay (Bio-Rad: Cat No. 5000006) before adjustment. The solution was incubated with Dynabeads Protein G (Invitrogen: Cat No. 10004D) conjugated with mouse monoclonal anti-FLAG M2 (Sigma: Cat No. F-3165) for 2 hr with slow rotation at 4°C, followed by washing with TBS with 0.1% Tween 20. The immunoprecipitates were eluted with 2x SDS sample buffer (100 mM Tris-HCl at pH 6.8, 4% SDS, 0.02% Bromophenol Blue, 20% glycerol, 2% 2-mercaptoethanol, 1% proteinase inhibitor cocktail) by boiling for 10 min. Each immunoprecipitate was then separated into two aliquots, treated with either water or Endo-H for 10 min at 37°C. Immunoblot analysis was performed using mouse monoclonal anti-FLAG M2 (Sigma: Cat No. F-3165; 1:5,000) antibody as primary antibody, and horseradish peroxidase-conjugated goat anti-mouse IgG (GE Healthcare: Cat No. NA931VS; 1:50,000) as secondary antibody. For loading control, immunoblot was performed using mouse anti-tubulin (Millipore Sigma: Cat No. MABT205; 1:5000). The protein blots were visualized using Chemiluminescence assay kit (Thermo Scientific: Cat No. 34095).

## Confocal microscopy and image analysis

Confocal microscopy images were taken on the Leica SP5X-WLL inverted confocal microscope (Solms, Germany). Time-lapse imaging of ERL1-YFP true leaves was prepared as described previously (*Peterson and Torii, 2012*). For internalization imaging of ERL1-YFP, ERL1ΔK-CFP and all other membrane organelle markers, maximum intensity projection of Z-stack images (0.33 µm step) covering the entire meristemoids were generated and subjected to analysis. The imaging was done with a 63x/1.2 W Corr lens on Leica SP5X. A 514 nm laser was used to excite YFP and emission window of 518–600 nm was used to collect YFP signal. A 458 nm laser was used to excite CFP and emission window of 470–510 nm was used to collect CFP signal. For the multicolor images of YFP and RFP, true leaves of 7-day-old transgenic seedlings were observed with a 63x/1.2 W Corr lens on Leica SP5X-WLL (Leica). 514 nm laser was used to excite YFP and 555 nm laser was used to excite RFP and FM4-64. Emission filter was set as 518 nm-550nm for YFP and 573–630 for RFP and FM4-64. Each experiment was repeated at least three times, each with multiple seedlings. Leica TCS-SP8 gSTED (Leica) was used for assessing the effects of Concanamycin A and high concentration of BFA on membrane organelle. For these purposes, images were taken using 93x glycerol immersion

objective lens and 488/500–530 nm, 514/518–550 nm, 561/600–650 nm for GFP, YFP, and RFP signals, respectively with Time Gating of 0.3–0.6 nm to eliminate chloroplast autofluorescence. The Leica LAS AF software (http://www.leica-microsystems.com), Fiji ( https://imagej.net/Fiji), and Imaris 8.1 (Bitplane) were used for post-acquisition image processing. Quantitative analysis of BFA signal intensity was performed using the Voxel counter plugin in Fiji briefly as the following. An extensive series of Z-stack confocal images covering the entire meristemoids were converted to 8-bit. Cut-off threshold was set from 35 to 200, which effectively removed objects with non-specific signals. An ROI was defined to cover the entire inside volume of each cell. Threshold sum (BFA bodies) and ROI sum (cell volume) for each ROI were subsequently measured. For Wm bodies quantification, z-stack confocal microscopy images were taken on the Leica SP5x inverted confocal microscope. The maximum projection of z-stack images was then generated by FIJI, and the number of detectable endosomes per cell or the number of Wm bodies (diameter >0.5 µm) were counted and recorded.

### Fluorescence recovery after photobleaching (FRAP) analysis

The FRAP experiments were conducted on ERL1-YFP using a 63x/1.2 W Corr lens on the Leica SP5X confocal microscope by photobleaching ~10% of the plasma membrane with 100% 405 nm laser power. 514 nm laser was used to excite YFP and emission window of 518–600 was used to collect YFP signal. Recovery of fluorescence was monitored in the photobleached plasma membrane for 6 min with 3 s intervals. A non-photobleached region was monitored meanwhile as an internal control. Average intensities of the region of interest were quantified with Leica LAS AF software. The exported data were analyzed and modeled by using the R-based FrapBot software (www.frapbot. kohze.com) (*Kohze et al., 2017*) with some modification to run on the local lab computer. The FRAP recovery curves were fitted to a single-parameter exponential model to determine the half time.

### Data plots and statistics

Graphs were generated using R (ver 3.4.1) using ggplot2. For box plots and violin plots, individual data points are plotted as dot plots, and for large sample numbers the dot plots were jittered with a position of 0.2. Welch's Two Sample T-Test and Person's correlation were analyzed using R. For specific codes, see Supplementary Source Code.

## Acknowledgements

We thank Prof. Takashi Ueda for GFP-SYP43, ARA6-GFP, RFP-ARA7, SYP43-RFP, and SYP22-RFP lines; Prof. Gian Pietro Di Sansebastiano for ST-RFP and KDEL-RFP constructs; Prof. Hugo Zheng for N-ST-YFP and N-YFP-HDEL constructs; Prof. Ikuko Hara-Nishimura for GFP-HDEL and ST-YFP lines; Prof. Joanne Chory for *BRI1pro::BRI1-GFP* line; Dr. Jin Suk Lee for pBAD::MEPFL6-6xHis plasmid (pJSL79); Geovanny Zarceno and Alex Hofstetter for technical assistance of peptide purification; Dr. Ayami Nakagawa for assistance in bioassays; Prof. Jenny Russinova for providing ES9-17 and insightful suggestions on experimental designs; Drs. Naoyuki Uchida, Ayami Nakagawa, Arvid Herrmann and Soon-Ki Han for commenting on the manuscript. A part of this work was conducted at Nagoya University Live Imaging Center supported by Japan Advanced Plant Science Research Network. This work was supported by the Gordon and Betty Moore Foundation (GBMF3035) to K.U.T., who is a Howard Hughes Medical Institute Investigator. K.U.T. acknowledges the generous support from The University of Texas at Austin as the Johnson and Johnson Centennial Chair.

## Additional information

### Funding

| Funder | Grant reference number | Author |
| --- | --- | --- |
| Howard Hughes Medical Institute | TORII | Keiko U Torii |
| Gordon and Betty Moore Foundation | GBMF3035 | Keiko U Torii |

University of Texas at Austin    Johnson & Johnson Centennial Chair of Plant Cell Biology    Keiko U Torii

The funders had no role in study design, data collection and interpretation, or the decision to submit the work for publication.

## Author contributions

Xingyun Qi, Data curation, Formal analysis, Validation, Investigation, Visualization, Methodology, Writing - original draft, Writing - review and editing; Akira Yoshinari, Resources, Data curation, Formal analysis, Validation, Investigation, Visualization, Methodology, Writing - original draft, Writing - review and editing; Pengfei Bai, Methodology, Writing - review and editing; Michal Maes, Resources, Writing - review and editing; Scott M Zeng, Software, Writing - review and editing; Keiko U Torii, Conceptualization, Resources, Data curation, Formal analysis, Supervision, Funding acquisition, Validation, Investigation, Visualization, Methodology, Writing - original draft, Project administration, Writing - review and editing

## Author ORCIDs

Xingyun Qi [iD] https://orcid.org/0000-0003-1261-1362
Akira Yoshinari [iD] https://orcid.org/0000-0002-7083-5674
Pengfei Bai [iD] https://orcid.org/0000-0002-2281-2881
Michal Maes [iD] https://orcid.org/0000-0002-7406-3982
Scott M Zeng [iD] https://orcid.org/0000-0003-3146-8207
Keiko U Torii [iD] https://orcid.org/0000-0002-6168-427X

## Decision letter and Author response

Decision letter https://doi.org/10.7554/eLife.58097.sa1
Author response https://doi.org/10.7554/eLife.58097.sa2

# Additional files

## Supplementary files

- Source code 1. R-scripts generated and used for this study.
- Supplementary file 1. List of DNA primers and their sequence used in the study.
- Transparent reporting form

## Data availability

All data generated or analyzed during this study are included in the manuscript and supporting files. Source R codes are provided.

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
