## [Decision Letter]

**Acceptance summary:**

This work provides a better understanding of ligand-dependent control of intracellular receptor distribution. We are very pleased to publish this paper at *eLife* because it outlines how vesicle trafficking contributes to signal integration during stomata differentiation in Arabidopsis.

**Decision letter after peer review:**

[Editors’ note: the authors submitted for reconsideration following the decision after peer review. What follows is the decision letter after the first round of review.]

Thank you for submitting your work entitled "The manifold actions of signaling peptides on subcellular dynamics of a receptor specify stomatal cell fate" for consideration by *eLife*. Your article has been reviewed by a Senior Editor, a Reviewing Editor, and three reviewers. The following individual involved in review of your submission has agreed to reveal their identity: Grégory Vert (Reviewer #2).

While the reviewers showed interest in your cell biological analysis, they also discussed several shortcomings, which precludes its publication at *eLife* in its present form. On one hand, they ask you to improve the quantification and image quality throughout your manuscript. On the other hand, they request additional experiments, including the generation of new lines (e.g. kinase dead ERL1), to further support your conclusion. It is policy of *eLife* to avoid lengthy re-review processes with uncertain outcomes. Therefore, we decided to reject the paper at this stage. Please find the specific comments below.

Reviewer #1:

The authors use genetics, microscopy and pharmacological treatments to determine sub cellular dynamics of the receptor ERL1 upon perceiving distinct ligands in the stomatal lineage of Arabidopsis. Through pharmacological experiments they show that the receptor ligand pair goes to the MVB and the vacuole upon endocytosis and can distinguish activated receptor routes towards degradation from general receptor endocytosis. Furthermore, the authors propose an attractive hypothesis where the coreceptor TMM modulates internalization of ERL1 in a ligand-specific manner. The experiments are well controlled, described in detail both in terms of methodology and sample size and the paper is written clearly and with an easy to follow structure. I have several concerns that I would like to see addressed and a couple of suggestions that would help clarity and visualization purposes.

I would like to stress that I am not a cell biologist and not 100% familiar with the limitations of the pharmacological assays.

A) Major concerns:

1) The data clearly shows that upon ligand perception the ERL1 gets internalized, send to MVB and likely degraded in the vacuole. While it is extremely interesting that these dynamics get modulated by co-receptor and the specific ligand, to me the mechanistic proof that internalization is relevant is somewhat lacking. Particularly when considering that very little of the PM-localized receptor gets internalized (unlike for FSL2, which the authors also discuss). If endocytosis and MVB mediated degradation is relevant then I would expect a patterning and clustering phenotype upon treatment with an endocytosis inhibitor (ES9-17 or Tyr A23) or even with Wm that blocks MVBs. Even though these experiments are tricky and probably must be done in a pulsed set-up, I think the manuscript would profit from phenotypic analysis of the leaf epidermis upon blocking the aforementioned processes (particularly endocytosis).

2) I am wondering why the authors focused on mEPF1 only since they are having mEPF2 available and Lee et al., 2012 showed that mEPF2 also binds to ERL1. Since the two peptides have not quite the same genetic role (one seems to inhibit the stomatal lineage at earlier stages than the other) it would be interesting to see if mEPF2 treatments induce the same internalization and sub-cellular dynamics and whether EPF2-ERL1 internalization is TMM dependent, too.

3) I feel somewhat uneasy about the fact that the authors use BFA treatment to infer anything about "general recycling" of the receptor. In my understanding BFA bodies are stalled TGN vesicles, which can go either to the MVB-vacuole pathway or get recycled to the PM. Therefore the BFA treatments and resulting BFA bodies rather indicate general internalization/endocytosis rather than recycling. To show recycling the authors would need to perform BFA washout experiments, but this is likely not possible since so much PM signal persists. I would like to ask the authors to change recycling for endocytosis/internalization throughout the manuscript.

Along the same lines, Figure 7 shows the receptor recycling as something isolated on the upper right corner. In my opinion this also goes through the TGN and should be drawn accordingly. Everything gets internalized through endocytosis and form TGN/EE. Then they either cycle back to the PM (recycling) or go to LE/MVB to be targeted to the vacuole. Please change the model accordingly and indicate which steps are inhibited by BFA, Wm, and ES9-17 and Tyr A23 in Figure 7.

B) Concerns in Figures:

4) General comments regarding figures:

- I would like to suggest that for all the quantifications of cells with bodies (eg. Figure 1C,E; Figure 2F, Figure 3E, F etc.) to label the y-axis with "% (or n) of YFP-positive bodies/endosomes/etc.".

- In addition, I would appreciate that for all plots the jitter of the data is indicated much like for the violin plots. It is currently missing in Figure 1C, 1E, 1G, Figure 2G.

- Finally, all bar charts should be box plots or violin plots (Figure 3G; Figure 4 – Figure supplement 1 B and C)

5) Figure 1:

I don't quite get what the difference between ERL1 vs. SYP22 is compared to SYP22 vs. ERL1. Please elaborate in the legend.

6) Figure 2:

In panel A it is not clear that the leftmost π stainings are different images than the ones showing YFP channels. Please split phenotype and YFP pictures into panel A and panel B, respectively.

For the FRAP experiments, I wonder why there seems to be more recovery albeit slower recovery in tmm. Is this something that was observed in all 9 replicates? If yes, please quantify and discuss this data, too. If not, I would suggest to use a different representative fluorescence recovery curve for tmm. And please also elaborate on why WT vs. tmm was so unbalanced in terms of replicates (2G, 3 vs. 9 respectively).

7) Figure 3.

In Figure 3E, it is really hard to see the FM4-64 internalization. Maybe use a more focussed and enlarged field of view?

I think that Figure 3F and Figure 3G should be combined or at the very least should be both transformed into violin plots with jitter so that they can be compared. It would also be much easier to read if it indicated in the figure and not just the legend that one is ERL1-YFP bodies and the other is ERL1deltaK-YFP bodies.

Please indicate the n for all treatments in 3G in the figure legend and also show as jitter on a violin or box plot. Otherwise it is really hard to judge the quality of this data.

8) Figure 5.

Please specify exact p values in Figure 5E and 5F as you did in Figure 4E and 4F.

9) Figure 6.

In 6B and 6C it is really hard to see the lines indicating where the fluorescence intensity is measured.

In 6D I would suggest to choose other colors than black dark gray and light gray. Or bigger difference between dark grey and light grey.

C) Concerns in text:

10) subsection “ERL1 is internalized through multivesicular bodies to vacuolar pathway in stomatal meristemoids”.

What do you mean with differentiating meristemoids? Are not all meristemoids differentiating?

11) Subsection “TMM is required for the process of ERL1 endocytosis in true leaves”.

I would appreciate if these quantifications would be included as box plots with jitter in the Figure 2—figure supplement 2 so that we can see the n, the variation in the data etc.

12) Subsection “An antagonistic EPFL peptide, Stomagen, elicits retention of ERL1-YFP in the endoplasmic reticulum”.

For the Stomagen experiments the authors use the er erl1 erl2 triple mutant to remove potential redundancy. I am a little worried that for some of the EPF1 and EPF6l experiments the results would look different in the triple mutant as well. I would appreciate the authors discussing this issue here or in the Discussion section.

13) Subsection “An antagonistic EPFL peptide, Stomagen, elicits retention of ERL1-YFP in the endoplasmic reticulum”.

I think that the data does not warrant such a strong conclusion. The ER accumulation is seen independent of TMM but might be stronger in tmm background. But for this statement to be valid the authors have to compare fluorescence intensity in wt vs. tmm background.

14) Subsection “Pharmacological treatment”.

I am a bit worried by the immensely different incubation times for MEPF1 / MEPFL6 compared to Stomagen (10minutes vs. 1hour). Would we also see ER retention if MEPF1 was incubated for 1h, too? Please show that this is not the case.

Reviewer #2:

The manuscript by Qi et al., reports on the characterization of the dynamics of the ERECTA-LIKE1 receptor and its dependence on the building of a functional receptor complex. It is a very interesting manuscript that provides a detailed analysis of ERL1 trafficking upon ligand binding or activation. Below are listed the most important points that need to be addressed by authors to improve the quality and readability of their contribution.

1) The confocal microscopy images require some more work to fully convince the reader :

- Figure 2A, rather than quantifying the ratio of cells showing ERL1 positive endosomes in WT and tmm, it would be more informative to quantify the number of ERL1-positive endosomes per cells in WT and tmm. To do so, z-stacks across the whole cell and max projection must be performed to avoid bias from uneven distribution of endosomes between, cell surface and middle section.

This comment applies to other figures. For example, the authors quantified the number of endosomes in a given confocal section in subsequent figures (Figure 5A) but must provide quantification per cell using z-stacks and max projection.

- Quantification of % of cells showing BFA or Wm bodies is not appropriate. Please also quantify BFA body number and size per cell. This is a better readout of endocytic defects.

- Some of the images would deserve higher magnification or enlarged panels to better visualize endosomes, lack of endosomes, or BFA/Wm bodies.

2) BFA has been widely characterized and used in Arabidopsis roots where it yields bodies containing aggregated TGN surrounded by Golgi. In aerial parts, BFA is known to trigger the collapsing of Golgi into the ER. Whether it also yields endosomal defects in stomatal meristemoids must be documented or tested if undocumented. The images shown the effect of BFA being very zoomed out, it is hard to evaluate the effect of BFA.

3) BFA leads, according to the authors, to the aggregation of TGN indicating that proteins trafficking through the TGN will end up (recycled or not) in BFA bodies. The fact that ERL1 is observed in BFA bodies in tmm upon CHX is therefore puzzling. One would anticipate a decrease (or even absence) of ERL1 if internalization is compromised. A better quantification of BFA bodies (BFA body number/cell and size; see 2)) is required to better characterize possible defects in ERL1 internalization. If the result stands, the authors will have to provide some explanations in the Discussion section. Regardless, a more direct investigation of ERL1 internalization kinetics in WT or tmm background by VAEM imaging would GREATLY strengthen the authors conclusions. I am not sure the authors have access to VAEM, but if so, I strongly encourage them to try.

4) The FRAP experiment raises some questions. First, the recovery of ERL1-YFP in erl1 is almost absent in my opinion. There is hardly any recovery compared to T0 (slope is flat). As such, it is hard to interpret and one may come up with a very different conclusion where ERL1 is largely immobile at the cell surface, and that it is more mobile in tmm. The authors should (i) show the prebleach intensity on their plot and (ii) image faster after photobleaching to catch the initial recovery phase better.

5) The internalization of ERL1 and targeting to MVB suggests that ERL1 is routed to the vacuole for degradation, which the authors have not shown. The authors must therefore assess ERL1 degradation by western blot if ERL1 is detectable in leaf extracts and/or by confocal imaging inhibiting the lytic activity of the vacuole (dark conditions of concanamycin A).

6) The ER retention of ERL1 upon stomagen application is puzzling yet very interesting. The authors must back up their observations by monitoring the endogenous ERL1 protein by immunofluorescence (or alternatively an ERL1-HA or FLAG if the authors do not have an ERL1 antibody) to ascertain that the YFP fusion protein, although active, represents the behavior of genuine ERL1 for ER retention. Second, the authors must address whether this is specific or ERL1 by testing the influence of stomagen on other secreted proteins (PM proteins) that woumd be expressed (e,dogenously or artificially) in stomata.

Reviewer #3:

This manuscript by Qi et al. investigated how the ERL1 receptor interprets different signal inputs and how this affects its subcellular dynamics. Overall, the manuscript is rather descriptive and fails to provide convincing mechanistic insight, leaving many loose ends. In addition, the text needs to be streamlined with respect to terminology (MVBs, endosomes, BFA-bodies, etc) and what is actually shown on the figures to avoid mis- or over-interpretation. Furthermore, the conclusions need to be more accurately phrased.

1) Regarding the statement that "TMM is required for ERL1 endocytosis", I would phrase this differently. TMM is required to form an active signaling complex, and in the absence of such an active signaling complex, no endocytosis occurs (as is also the case upon Stomagen treatment or when the kinase domain is removed). How do the authors explain the Stomagen-mediated accumulation of ERL1 in the endoplasmic reticulum? This must be a different pool of ERL1 distinct from the Stomagen-bound ERL1. Do the authors assume a regulatory mechanism whereby Stomagen-bound ERL1 signals to control the production and ER retention of new ERL1? In this context, I would conclude that it is not inefficient ERL1 endocytosis that leads to retention in the ER, but an inactive receptor that stays on the membrane which leads to no new secretion. Does the ERL1Δkinase also accumulate in the ER?

2) With the current data, it cannot be concluded that kinase activity is required for ERL1 internalization. There could be internalization motifs in the kinase domain. To conclusively demonstrate this, an ERL1 kinase dead variant needs to be analyzed with respect to internalization. In the absence of TMM, un-activated ERL1 receptors are not readily targeted for endocytosis and remain stable at the membrane. What happens with ERL1Δkinase endocytosis in the presence of ligand?

3) The authors use different markers to reveal the localization of ERL1-YFP. However, a typical experiment to study endocytosis is the co-localization with FM4-64. Figure 2—figure supplement 2A and Figure 3E provide some idea, but the individual endosomes should be tracked. BFA bodies and MVBs are a proxy to study this, but then the phrasing in the text should be accordingly.

4) TMM is required for endocytosis of ERL1, and not recycling. BFA blocks the secretion of ERL1. One would expect that if there is less endocytosis, there is also less ERL1 in BFA bodies. However, based on the data in Figure 2 this is not the case. Also, Figure 2B shows an increased ERL1-YFP intensity upon BFA treatment in tmm, so both the number of cells with BFA bodies and the signal intensity would be meaningful to report. I would rephrase the conclusion in subsection “TMM is required for the process of ERL1 endocytosis in true leaves”: the absence of tmm does not impact the ERL1 secretory pathway, as reflected in the presence of ERL1-YFP positive BFA bodies.

5) The authors invoke that distinct EPF/EPFL peptide ligands activate a sub-population of ERL1 receptor complex to internalize through a TMM-based discriminatory system. This seems independent of distinct expression domains of these peptides as purified EPFL6 does not have the same effect as EPF1. The authors suggest distinct localization in nanodomains, but do not investigate this. It would be valuable to see FRAP analyses on ERL1-YFP +/1 EPF1 or EPFL6 in erl1 and erl1 tmm.

6) Throughout the manuscript, MVBs should be used when describing the "large dots" on the figures. I am not sure if the resolution allows to see endosomes. Are Figure 4E-F and Figure 5E-F really showing number of endosomes or should this be MVBs?

7) The authors write "TMM is required for endocytic sorting of ERL1 to MVBs, which is a hallmark for receptor degradation". I guess that given the fact that in tmm ERL1-YFP signal increases in the ER, it is rather difficult to investigate this statement using e.g. Western Blot.

8) Given the recent knowledge on Tyr A23, I would only include the more specific ES9-17. Is this effect on ES9-17 specific for ERL1-YFP, or are other receptor proteins also accumulating in the ER?

[Editors’ note: further revisions were suggested prior to acceptance, as described below.]

Thank you for submitting your article "The manifold actions of signaling peptides on subcellular dynamics of a receptor specify stomatal cell fate" for consideration by *eLife*. Your article has been reviewed by Christian Hardtke as the Senior Editor, Jürgen Kleine-Vehn as the Reviewing Editor, and two reviewers. The reviewers have opted to remain anonymous.

The reviewers have discussed the reviews with one another and the Reviewing Editor has drafted this decision to help you prepare a revised submission.

Your manuscript reveals remarkable insights into peptide ligands and how they modulate the intracellular distribution of their receptors. We invite you to resubmit your paper for final acceptance, considering the following main suggestions of the reviewers:

I) The reviewers ask you to clarify several issues, such as statistical evaluation and quantifications, which may not require novel experiments.

II) During their discussion, the reviewers defined a better characterization and integration of the ERL1Δkinase line, as an essential part of the revision, and one that unfortunately requires additional experimentation.

III) The reviewers judge that some experiments aimed at a better integration of your data into developmental context would substantially strengthen the impact of your manuscript. On the other hand, they do not believe that such an insight is essential for acceptance of the manuscript.

Please, see the detailed comments of the reviewers below.

Reviewer #1:

This manuscript by Qi et al., investigated how the ERL1 receptor interprets different signal inputs and how this affects its subcellular dynamics. There are several novel and interesting insights, but still some loose ends remain. The authors have addressed several of the comments raised previously and improved the manuscript.

Reviewer #2:

The authors undertook a remarkable effort to improve the cell biological aspects of this manuscript and I do appreciate the new experiments and careful quantification of the BFA experiments and analysis of new marker lines with expert Dr. Akira Yoshinari. In addition, the ConcavA treatments and the clear visualization of ERL1 in vacuoles adds significantly to this manuscript. However, I would like to stress two of my previous concerns again.

1) Previous point 1:

From a mechanistic/genetic perspective, the functional relevance of ERL1 endocytosis and targeting to MVB/LE and vacuole is still missing. I do completely understand that a phenotypic assay is not doable due to long-time cytotoxic effects of BFA or Wm. I also agree that strong genetic support can only come through analysis of how ERL1 is modified (phosphorylation etc) and how these modifications affect endocytosis, but this is clearly out of scope of this article. Nevertheless, in Qi et al., 2017, the authors show that the EPF1-ERL1 module directly represses MUTE in an autocrine manner. When EPF1 expression is induced, a clear decrease in MUTE-YFP fluorescence can be observed already 15hours post induction. I wonder whether induction of EPF1 together with mock or toxin treatment (Wm?) in the first couple of hours could reveal differences in how efficiently MUTE-YFP is downregulated. This could give at least somewhat functional support to the hypothesis that the endocytosis and targeting to the vacuole is required for efficient downregulation of key targets by the EFP1-ERL1 module.

2) Previous point 12:

Even though the authors stated in their response that they elaborated on ER ERL1 and ERL2 redundancy in the discussion, I cannot find new information or statements regarding the fact that the Stomagen-induced ER-retainment of ERL1 is much stronger in the triple erecta-erl1-erl2 background. I am aware that this is a subtle difference but as the authors confirm an important one. So please elaborate on this in the results (line 298ff) and clearly mention that Stomagen has been shown to interact initially with ER before acting through ERL1 and ERL2 at later stages, which is why a triple mutant is needed here and ER retainment in the single erl1 is much weaker.

---

## [Author Response]

We took the constructive criticisms from the three Peer Reviewers very seriously, and performed the following experiments to fully address their concerns.

i) Concanamycin A treatment to seedlings expressing ERL1-YFP: The results demonstrate that ERL1-YFP is destined to a vacuole (see Revised Figure 2).

ii) Stomagen application to BRI1pro::BRI1-GFP seedlings: BRI1-GFP is endongenously expressed in leaf epidermis, includingstomatal meristemoids. The results show that Stomagen peptide does not influence subcellular localization of BRI1, thereby supporting the notion that the effects of Stomagen on ERL1 is specific (see Revised Figure 8–figure supplement 2).

iii) Extensive documentation of BFA body formation in stomatal meristemoids: We used eight different endomembrane organelle markers with an endocytic tracer dye FM4-64 to fully document the behavior of endomembrane system upon BFA treatment in meristemoids. Our results clearly demonstrate that, at low concentration (30 microM), BFA body formation occurs just like in root cells (See Revised Figure 3 and Figure 3–figure supplement 1).

Furthermore, we re-analyzed BFA and Wm treatment data as suggested by the Reviewers, and provided enlarged, high-resolution confocal images as much as possible. By analyzing the number of endosomes/BFA/Wm bodies per cell as well as the volume ratio, we were able to support our original conclusion that TMM is critical for the internalization of EPF1-perceived ERL1. Please see our point-by-point response for specifics.

Reviewer #1:[…] A) Major concerns:1) The data clearly shows that upon ligand perception the ERL1 gets internalized, send to MVB and likely degraded in the vacuole. While it is extremely interesting that these dynamics get modulated by co-receptor and the specific ligand, to me the mechanistic proof that internalization is relevant is somewhat lacking. Particularly when considering that very little of the PM-localized receptor gets internalized (unlike for FSL2, which the authors also discuss). If endocytosis and MVB mediated degradation is relevant then I would expect a patterning and clustering phenotype upon treatment with an endocytosis inhibitor (ES9-17 or Tyr A23) or even with Wm that blocks MVBs. Even though these experiments are tricky and probably must be done in a pulsed set-up, I think the manuscript would profit from phenotypic analysis of the leaf epidermis upon blocking the aforementioned processes (particularly endocytosis).

We thank reviewer 1 for finding our work extremely interesting. We agree with reviewer 1 that it would be ideal if we could use endocytosis inhibitors to block ERL1 endocytosis and then characterize the resulting stomatal pattering. Unfortunately, extended treatment of these inhibitors for several days to observe stomatal development is not possible due to high cytotoxicity of these inhibitors. All of our results clearly show the subcellular dynamics of ERL1 upon perceiving its ligands with different functions (EPF1, Stomagen, and EPFL6). Furthermore, the dominant-negative receptor that cannot signal is compromised in internalization. In the future, identification of receptor modification which influence internalization (e.g. phosphorylation, ubiquitination) may fully answer reviewer 1's question, which are out-of-scope of this manuscript.

2) I am wondering why the authors focused on mEPF1 only since they are having mEPF2 available and Lee et al., 2012 showed that mEPF2 also binds to ERL1. Since the two peptides have not quite the same genetic role (one seems to inhibit the stomatal lineage at earlier stages than the other) it would be interesting to see if mEPF2 treatments induce the same internalization and sub-cellular dynamics and whether EPF2-ERL1 internalization is TMM dependent, too.

We have previously reported that mEPF1 and ERL1 are both expressed in stomatal meristemoids and participate in autocrine signaling to restrict stomatal differentiation potential (Qi et al., 2017), in addition to their well-established role in enforcing the stomatal one-cell spacing rule (Hara et al., 2007). The clear expression pattern of ERL1 in the meristemoids and the known consequence of autocrine signaling make mEPF1-ERL1 a unique and powerful system to address how signal perception (activation vs inactivation) triggers receptor subcellular dynamics. While mEPF2 can also be perceived by ERL1, its major receptor, ERECTA is ubiquitously expressed at very high level (Lee et al., 2012), and we predict that this will interfere with the proper interpretation. In the revised manuscript, we emphasized our rationale in the Introduction.

3) I feel somewhat uneasy about the fact that the authors use BFA treatment to infer anything about "general recycling" of the receptor. In my understanding BFA bodies are stalled TGN vesicles, which can go either to the MVB-vacuole pathway or get recycled to the PM. Therefore the BFA treatments and resulting BFA bodies rather indicate general internalization/endocytosis rather than recycling. To show recycling the authors would need to perform BFA washout experiments, but this is likely not possible since so much PM signal persists. I would like to ask the authors to change recycling for endocytosis/internalization throughout the manuscript.Along the same lines, Figure 7 shows the receptor recycling as something isolated on the upper right corner. In my opinion this also goes through the TGN and should be drawn accordingly. Everything gets internalized through endocytosis and form TGN/EE. Then they either cycle back to the PM (recycling) or go to LE/MVB to be targeted to the vacuole. Please change the model accordingly and indicate which steps are inhibited by BFA, Wm, and ES9-17 and Tyr A23 in Figure 7.

We truly thank reviewer 1 for raising an important point. We agree that BFA treatment affects not only recycling but also internalization/endocytosis. As reviewer 1 mentions, "BFA washout" time course experiments is generally performed to measure the rate of recycling from BFA body to Plasma membrane. Given that our results indicate that the main effects of *tmm* mutation is reduced internalization/endocytosis, we do not believe that the additional BFA washout experment is necessary. In response to reviewer 1, we have changed recycling to recycling and endocytosis/internalization throughout the manuscript. Likewise, we have revised the model figure to correctly reflect the literature and our inhibitor results (now Figure 9).

During the revision process, we extensively discussed how to address the questions raised by the reviewers regarding our BFA treatment experiments. We invited an expert, Dr. Akira Yoshinari, who studies subcellular dynamics of polarly-localized membrane transporters. Through collaboration, we performed a series of BFA-treatment experiments in the context of stomatal meristemoids using a variety of subcellular membrane markers (revised Figure 3 and Figure 3—figure supplement 1; see our specific response to reviewer 2). Furthermore, we performed a pharmacological treatment using Concanamycin A, a specific inhibitor of vacuolar H^+^-ATPase that is known to reduce protein degradation in the lytic vacuole (e.g. Kleine-Vehn et al., 2008). We now demonstrate that ERL1 indeed accumulates in a vacuole (see revised Figure 2). In summary, all the additional experiments we performed support our original findings that ERL1, upon activated by EPF peptides, is subjected to endocytosis/internalization destined to vacuole and the process gets compromised in the absence of TMM.

B) Concerns in Figures:4) General comments regarding figures:- I would like to suggest that for all the quantifications of cells with bodies (eg. Figure 1C,E; Figure 2F, Figure 3E, F etc.) to label the y-axis with "% (or n) of YFP-positive bodies/endosomes/etc.".

We have modified the y-axis labels and re-analyzed our data to plot the number of

YFP positive bodies/endosomes/etc per cell (see revised Figure 4F, Figure 6—figure supplement 1B, C). For a quantitative analysis of BFA bodies, as also requested by reviewer 2, we have re-analyzed the voxel YFP intensity within the cell (representing endosomes and BFA bodies) per voxel YFP intensity of cell periphery (plasma membrane) and plotted the data to reveal subtle reduction in *tmm* mutant background (see revised Figure 4D). This analysis indeed revealed a reduction in the BFA body signals by the loss-of-function mutation in *TMM*. We additionally reanalyzed Wm treatment and plotted boxplots as number of Wm bodies per cell (see revised Figure 4F, Figure 4 —figure supplement 2D).

- In addition, I would appreciate that for all plots the jitter of the data is indicated much like for the violin plots. It is currently missing in Figure 1C, 1E, 1G, Figure 2G.

Provided for all box plot graphs (see revised Figure 1C, Figure 4D, 4F, Figure 6—figure supplement 1B, C).

- Finally, all bar charts should be box plots or violin plots (Figure 3G; Figure 4 – Figure supplement 1 B and C)

Done. The original Figure 3G is a percentage of all cell counted. We removed the chart and just indicated the percentage in the text.

5) Figure 1:I don't quite get what the difference between ERL1 vs. SYP22 is compared to SYP22 vs. ERL1. Please elaborate in the legend.

We thank reviewer 1 for asking for the clarification.

ERL1 vs. SYP43 indicates how many ERL1-YFP endosomes co-localize with SYP43-RFP. Co-localized endosomes are in white arrowheads.

SYP43 vs. ERL1 indicates how many SYP43-RFP endosomes co-localize with ERL1-YFP endosomes. Co-localized endosomes are in white arrowheads.

The % colocalization are not identical because some endosomes only accumulate ERL1-YFP whereas some other endosomes only accumulate SYP43-RFP. This disparity is most evident when comparing SYP22-RFP and ERL1-YFP, because almost all endosomes accumulates YFP signals alone, whereas nearly all of the RFP endosomes co-localize with YFP signals. To be clear to readers, we now describe this in the figure legend. Thank you.

6) Figure 2:In panel A it is not clear that the leftmost π stainings are different images than the ones showing YFP channels. Please split phenotype and YFP pictures into panel A and panel B, respectively.

Reviewer 1 is correct. Our intention in the original Figure was to show the stomatal cluster phenotype of *tmm*. To be unambiguous, we revised the figure to split the figure panels (now Figure 4A and B).

For the FRAP experiments, I wonder why there seems to be more recovery albeit slower recovery in tmm. Is this something that was observed in all 9 replicates? If yes, please quantify and discuss this data, too. If not, I would suggest to use a different representative fluorescence recovery curve for tmm. And please also elaborate on why WT vs. tmm was so unbalanced in terms of replicates (2G, 3 vs. 9 respectively).

We have used FRAP Bot program (www.frapbot.kohze.com) to normalize and fit our fluorescence recovery curve to a single-parameter exponential model. The fitted curve, but not the original fluorescence intensity changes, are provided in the graphs (revised Figure 4G). As far as our understanding, recovery curve is a standard way to determine the half time, and similar curve fitting has been done by many others in the field (e.g. Martinière et al., 2012; Zhang et al., 2016). We performed 10 independent FRAP experiments, and the FRAP Bot program chose the experiment that passed their criteria (we thought it would be better for the algorithm to perform unbiased curve fitting, rather than we choose the 'representative' data). To be clear, we indicated this in our revised Method section. Furthermore, in response to a similar critique from reviewer 2, we now provide a graph of observed florescence intensity change in the representative FRAP experiment (Figure 4—figure supplement 3 in the revision) as well as confocal images throughout a representative FRAP experiment (Video 2 and Video 3).

7) Figure 3.In Figure 3E, it is really hard to see the FM4-64 internalization. Maybe use a more focussed and enlarged field of view?

Thank you for the suggestion. As also requested by reviewer 2, we now provide enlarged field of view throughout our confocal microscopy image figures.

I think that Figure 3F and and Figure 3G should be combined or at the very least should be both transformed into violin plots with jitter so that they can be compared. It would also be much easier to read if it indicated in the figure and not just the legend that one is ERL1-YFP bodies and the other is ERL1deltaK-YFP bodies.Please indicate the n for all treatments in 3G in the figure legend and also show as jitter on a violin or box plot. Otherwise it is really hard to judge the quality of this data.

As suggested, we removed the Figure 3G, and instead, indicated the percentage of cells with endosome bodies within the manuscript.

8) Figure 5.Please specify exact p values in 5E and 5F as you did in 4E and 4F.

We have performed Welch’s Two-Sample T-Test and the exact p values for pairwise comparison are specified (now revised Figures 6E, 6F, Figure 7E, and 7F).

9) Figure 6.In 6B and 6C it is really hard to see the lines indicating where the fluorescence intensity is measured.

As suggested, we made the lines thicker (Figure 8 in the revision). Thank you.

In 6D I would suggest to choose other colors than black dark gray and light gray. Or bigger difference between dark grey and light grey.

Done (gray arrows blue and cyan now).

C) Concerns in text:10) subsection “ERL1 is internalized through multivesicular bodies to vacuolar pathway in stomatal meristemoids”.What do you mean with differentiating meristemoids? Are not all meristemoids differentiating?

Text changed to late meristemoids.

11) Subsection “TMM is required for the process of ERL1 endocytosis in true leaves”.I would appreciate if these quantifications would be included as box plots with jitter in the Figure 2—figure supplement 2 so that we can see the n, the variation in the data etc.

Thanks for the suggestion. As also requested by other two Reviewers, we have plotted the number of YFP+ BFA/Wm bodies per cell (per ERL1-YFP expressing meristmoid) and replotted as box plots with jitter (now Figure 4—figure supplement 2 in the revision).

12) Subsection “An antagonistic EPFL peptide, Stomagen, elicits retention of ERL1-YFP in the endoplasmic reticulum”.For the Stomagen experiments the authors use the er erl1 erl2 triple mutant to remove potential redundancy. I am a little worried that for some of the EPF1 and EPF6l experiments the results would look different in the triple mutant as well. I would appreciate the authors discussing this issue here or in the Discussion section.

We appreciate reviewer 1 for commenting on this, as we agree this is a valid point. Our group has extensively studied a redundancy among three ERECTA family genes and their specific interaction with TMM as well as different EPF peptides (e.g. Shpak, 2005; Hara, 2009; Lee, 2012; Lee, 2015); often these subtle yet important differences are ignored by the scientific community. Through these studies, we have found that whereas EPF1 has rather effects to activate ERL1, Stomagen acts stepwise throughout stomatal lineages, earlier with ERECTA, and ERL1 and ERL2 at a later stage). This complicates the interpretation. We elaborated this in the Discussion section.

13) Subsection “An antagonistic EPFL peptide, Stomagen, elicits retention of ERL1-YFP in the endoplasmic reticulum”.I think that the data does not warrant such a strong conclusion. The ER accumulation is seen independent of TMM but might be stronger in tmm background. But for this statement to be valid the authors have to compare fluorescence intensity in wt vs. tmm background.

We have changed the sentence to "suggesting that the absence of TMM intensifies the accumulation of ERL1 in the endoplasmic reticulum".

14) Subsection “Pharmacological treatment”.I am a bit worried by the immensely different incubation times for MEPF1 / MEPFL6 compared to Stomagen (10minutes vs. 1hour). Would we also see ER retention if MEPF1 was incubated for 1hour, too? Please show that this is not the case.

Due to very low amount of bioactive MEPF1 peptide we currently have, we did not perform additional time course experiments. Having said that, using very robust cheically inducible EPF1 overexpression system (which we used previously to robustly trigger EPF1 overexpression in time-controlled manner; Lee et al., 2012, Qi et al., 2017). We performed time course overexpression (which occurs with in less than 1 hour) up to 8 hours, and we did not observe any ERL1-YFP signal in the ER. Thus, it is highly unlikely that the different subcellular localization pattern of ERL1-YFP by MEPF1 or Stomagen application is owing to 50 min difference. If reviewer 1 request, we would be happy to provide the data as Supplemental Figure.

Reviewer #2:The manuscript by Qi et al., reports on the characterization of the dynamics of the ERECTA-LIKE1 receptor and its dependence on the building of a functional receptor complex. It is a very interesting manuscript that provides a detailed analysis of ERL1 trafficking upon ligand binding or activation. Below are listed the most important points that need to be addressed by authors to improve the quality and readability of their contribution.

We thank reviewer 2 for finding our manuscript very interesting.

1) The confocal microscopy images require some more work to fully convince the reader :- Figure 2A, rather than quantifying the ratio of cells showing ERL1 positive endosomes in WT and tmm, it would be more informative to quantify the number of ERL1-positive endosomes per cells in WT and tmm. To do so, z-stacks across the whole cell and max projection must be performed to avoid bias from uneven distribution of endosomes between, cell surface and middle section.This comment applies to other figures. For example, the authors quantified the number of endosomes in a given confocal section in subsequent figures (Figure 5A) but must provide quantification per cell using z-stacks and max projection.

We sincerely appreciate reviewer 2's expert suggestions. We would like to emphasize that all of our quantitative analyses of ERL1-YFP endocytosis were performed by using maximum intensity projection of Z-stack images covering the entire meristmoids, not using single-plane images. We clarified this in the Materials and methods section.

- Quantification of % of cells showing BFA or Wm bodies is not appropriate. Please also quantify BFA body number and size per cell. This is a better readout of endocytic defects.

As suggested by reviewer 2 and all other reviewers, we re-analyzed our images and quantified BFA or Wm body number per cell (revised Figure 4D, F, Figure 4 —figure supplement1D, E, Figure 6—figure supplement 1B, C) and plotted as boxplots with individual data points as jitter.

For comparing the BFA-body formation between WT vs. *tmm*, in order to see subtle differences, we additionally performed quantification of the voxel signal intensity of inside of the cell (corresponding to endosomes/BFA bodies) per plasma membrane. This analysis was indeed very informative, as we were able to show a subtle yet statistically different voxel signal intensity ratio between WT and *tmm* (revised Figure 4D). It is well known that BFA strongly inhibits recycling, but also interferes with endocytosis/internalization. The subtle yet significant reduction in BFA bodies in *tmm* supports our original conclusion that ERL1 internalization requires its co-receptor TMM. Thank you for the great suggestion!

- Some of the images would deserve higher magnification or enlarged panels to better visualize endosomes, lack of endosomes, or BFA/Wm bodies.

As suggested, we now provide enlarged panels throughout (e.g. revised Figure 1A, 1D, Figure 4E, Figure 5D).

2) BFA has been widely characterized and used in Arabidopsis roots where it yields bodies containing aggregated TGN surrounded by Golgi. In aerial parts, BFA is known to trigger the collapsing of Golgi into the ER. Whether it also yields endosomal defects in stomatal meristemoids must be documented or tested if undocumented.

We thank reviewer 2 for bringing up the previous knowledge to our attention.

Indeed, Robinson et al., (2006) and Langhans et al., (2011) reported that BFA triggers re-absorption of Golgi membrane to the ER in leaves. These authors used rather high concentrations of BFA (50 µM for the former report and 90, 180, and 360 µM for the latter report) when observing the collapse of Golgi to ER. By contrast, we used low BFA concentration (30 µM), a condition generally used to block recycling/endocytosis in roots.

Since we did not find previous literatures that study membrane organelle dynamics in BFA-treated stomatal meristemoids, we took this revision process as an opportunity for the extensive documentations. For this purpose, we collaborated with Dr. Akira Yoshinari, an expert in subcellular dynamics of membrane transporters in plants (Dr. Yoshinari is the second author of our revised manuscript).

We collected eight different membrane organelle markers: GFP-HDEL (ER); ST-

YFP (Golgi); YFP-SYP32 (cis-Golgi); GFP-SYP43 (TGN/EE); YFP-RabA1e (TGN, PM); YFP-RabA5d (uncharacterized endosomes); ARA6-GFP (MVB/late endosomes, PM); YFP-ARA7 (MVB/late endosomes), and co-stained with the endocytosis tracer FM4-64, which initially stains plasma membrane.

First, we treated GFP-HDEL/FM4-64 and ST-YFP/FM4-64 with BFA at different concentrations (0, 30, 90, 180 µM). Treatment of low BFA concentration at 30 µM (which we used throughout our study) does not confer any discernable effects on ER (GFP-HEDL, revised Figure 3A, B). Under this condition, BFA bodies are surrounded by Golgi marker ST-YFP (Figure 3C), just as expected for the BFA treatment in roots (Figure 3D) (Langhans et al., 2011). In contrast, treatment of BFA at very high concentration, 180 µM, conferred aberrant spherical structure of ER (Figure 3A, B), which was also previously reported (Nakano et al., 2009). Moreover, we observed Golgi absorbed ER, which confers ring-like structure of membrane accumulating ST-YFP (Figure 3C). In short, we recapitulated the previous observations of collapsing of Golgi into the ER only at extremely high BFA concentration at 180 µM.

Next, we investigated the intactness of different endomenbrane organelles in stomatal meristemoids when treated in 30 µM BFA. For this purpose, seven additional endomembrane markers were carefully examined (YFP-SYP32, N-STYFP, GFP-SYP43, YFP-RabA1e, YFP-RabA5d, ARA6-GFP and YFP-ARA7 costained with FM4-64. All marker lines show the formation of characteristic BFA bodies in the meristemoids, without any collapse of Golgi into ER (Figure 3—figure supplement 1).

Taken together, we conclude that BFA treatment at 30 µM, in our hands, confers the formation of characteristic BFA bodies in the stomatal meristemoids in leaf, and thereby supporting the validity of our BFA treatment to study ERL1 subcellular dynamics. We strongly believe that our extensive documentation will serve as a guide to the field, and we thank reviewer 2 again for the excellent advice.

The images shown the effect of BFA being very zoomed out, it is hard to evaluate the effect of BFA.

As also requested by reviewers 1 and 3, we now provide enlarged field of view throughout our confocal microscopy image figures whenever possible.

3) BFA leads, according to the authors, to the aggregation of TGN indicating that proteins trafficking through the TGN will end up (recycled or not) in BFA bodies. The fact that ERL1 is observed in BFA bodies in tmm upon CHX is therefore puzzling. One would anticipate a decrease (or even absence) of ERL1 if internalization is compromised. A better quantification of BFA bodies (BFA body number/cell and size; see 2)) is required to better characterize possible defects in ERL1 internalization. If the result stands, the authors will have to provide some explanations in the Discussion section. Regardless, a more direct investigation of ERL1 internalization kinetics in WT or tmm background by VAEM imaging would GREATLY strengthen the authors conclusions. I am not sure the authors have access to VAEM, but if so, I strongly encourage them to try.

We have re-analyzed the effects of BFA on ERL1-YFP in WT vs. *tmm* backgrounds by quantifying the ratio of YFP+ signals within the cell (voxel) per YFP+ signals at plasma membrane using our maximal intensity projection of Z-stack images (formerly Figure 2C, now Figure 4D). The analysis revealed a statistically significant reduction of ERL1-YFP in BFA bodies in *tmm* mutant, but it is not completely absent. These results suggest that in the absence of TMM, ERL1-YFP internalization is compromised but not absent. This is most likely due to the presence of EPF-LIKE peptide family members that do not require TMM (e.g. EPFL4 and EPFL6). We discussed this point in the Discussion. We agree that VAEM (or TIRF) microscopy would be an interesting next step to study receptor internalization dynamics in the future.

4) The FRAP experiment raises some questions. First, the recovery of ERL1-YFP in erl1 is almost absent in my opinion. There is hardly any recovery compared to T0 (slope is flat). As such, it is hard to interpret and one may come up with a very different conclusion where ERL1 is largely immobile at the cell surface, and that it is more mobile in tmm. The authors should (i) show the prebleach intensity on their plot and (ii) image faster after photobleaching to catch the initial recovery phase better.

We have used FRAP Bot program (www.frapbot.kohze.com) to normalize and fit our fluorescence recovery curve to a single-parameter exponential model. The fitted curve, but not the original fluorescence intensity changes, are provided in the original graphs. As far as our understanding, recovery curve is a standard way to determine the half time, and similar curve fitting has been done by many others in the field (e.g. Martinière et al., 2012; Zhang et al., 2016).

As requested by reviewer 2, we are providing both a graph of observed prebleach florescence intensity and intensity recovery in representative FRAP experiments (Figure 4—figure supplement 3 in the revision) as well as confocal images throughout a representative FRAP experiment (Video 2 and Video 3).

5) The internalization of ERL1 and targeting to MVB suggests that ERL1 is routed to the vacuole for degradation, which the authors have not shown. The authors must therefore assess ERL1 degradation by western blot if ERL1 is detectable in leaf extracts and/or by confocal imaging inhibiting the lytic activity of the vacuole (dark conditions of concanamycin A).

We thank reviewer 2 for the excellent suggestion. With Dr. Yoshinari, we performed Concancmycin A treatment of ERL1-YFP seedlings that were pretreated with FM4-64 following the published protocols (e.g. Kleine-Vehn et al., 2008). Indeed, we observed ERL1-YFP accumulation in the vacuole of stomatal meristemoids (see revised Figure 2). The data support our original conclusion that ERL1, upon activated by EPF peptides, is subjected to internalization and targeted to MVP and destined to the vacuole.

6) The ER retention of ERL1 upon stomagen application is puzzling yet very interesting. The authors must back up their observations by monitoring the endogenous ERL1 protein by immunofluorescence (or alternatively an ERL1-HA or FLAG if the authors do not have an ERL1 antibody) to ascertain that the YFP fusion protein, although active, represents the behavior of genuine ERL1 for ER retention. Second, the authors must address whether this is specific or ERL1 by testing the influence of stomagen on other secreted proteins (PM proteins) that woumd be expressed (e,dogenously or artificially) in stomata.

We do not have endogenous antibody that specifically detect ERL1, and immunostain of epidermal cells have been unsuccessful in our hands. Our Stomagen treatment experiments have been performed multiple times, not just by Confocal microscopy observations but also using biochemical Endo-H treatment.

As suggested by reviewer 2, we treated Stomagen peptide on Arabidopsis seedlings expressing BRI1-GFP driven by its endogenous promoter (*BRI1pro::BRI1-GFP*). As reported previously (Friedrichsen et al., 2000), BRI1 is ubiquitously expressed in the epidermis, including meristemoids and stomata (see Figure 8—figure supplement 3). The application of Stomagen did not change plasma-membrane localization of BRI1. We did not detect BRI1-GFP in the ER, indicating that the influence of Stomagen on the ER retention is not universal to LRR-RLK.

Reviewer #3:This manuscript by Qi et al. investigated how the ERL1 receptor interprets different signal inputs and how this affects its subcellular dynamics. Overall, the manuscript is rather descriptive and fails to provide convincing mechanistic insight, leaving many loose ends. In addition, the text needs to be streamlined with respect to terminology (MVBs, endosomes, BFA-bodies, etc) and what is actually shown on the figures to avoid mis- or over-interpretation. Furthermore, the conclusions need to be more accurately phrased.

We sincerely appreciate reviewer 3's expert comments. Our work reports extensive documentation of receptor subcellular localization upon perceiving signaling ligands with different activities as well as when the receptor cannot signal, using the system in which one cell type (stomatal meristemoids) respond to multiple different, and conflicting signals. Thus, we believe our finding add new insight and useful knowledge (together with extensive study of BFA treatment in the meristemoids now presented in Figure 3) to the field. We agree with reviewer 3, and we took our best effort to accurately phrase the observations and results in the revision.

1) Regarding the statement that "TMM is required for ERL1 endocytosis", I would phrase this differently. TMM is required to form an active signaling complex, and in the absence of such an active signaling complex, no endocytosis occurs (as is also the case upon Stomagen treatment or when the kinase domain is removed). How do the authors explain the Stomagen-mediated accumulation of ERL1 in the endoplasmic reticulum? This must be a different pool of ERL1 distinct from the Stomagen-bound ERL1. Do the authors assume a regulatory mechanism whereby Stomagen-bound ERL1 signals to control the production and ER retention of new ERL1? In this context, I would conclude that it is not inefficient ERL1 endocytosis that leads to retention in the ER, but an inactive receptor that stays on the membrane which leads to no new secretion. Does the ERL1Δkinase also accumulate in the ER?

We agree with reviewer 3 that TMM is required to form an active signaling complex, and in the absence of the active signaling complex endocytosis of the receptor gets compromised. We have revised our text throughout the manuscript to be clear. As for the effects of Stomagen on ERL1 accumulation in the ER, together with the pharmacological studies using TyrA23 and ES9-17, suggests that reduced signal activation by Stomagen will result in stalling of ERL1 in the ER. We wish to study the exact subcellular mechanism of the observed stalling in the future.

2) With the current data, it cannot be concluded that kinase activity is required for ERL1 internalization. There could be internalization motifs in the kinase domain. To conclusively demonstrate this, an ERL1 kinase dead variant needs to be analyzed with respect to internalization. In the absence of TMM, un-activated ERL1 receptors are not readily targeted for endocytosis and remain stable at the membrane. What happens with ERL1Δkinase endocytosis in the presence of ligand?

We fully concur with reviewer 3. We have no intention to claim that the kinase activity is required for the internalization. We are fully aware of the fact that removal of the entire cytoplasmic domain will remove potential binding interface with endocytosis adaptor proteins, including those in AP complex or TPLATE complex. Our intention here is to show that a dominant-negative receptor, which is known to severely interfere with it signal transduction pathway (Shpak et al., 2003; Lee et al., 2012), is largely compromised in internalization. We agree that to examine whether the kinase activity affect internalization, kinase-dead variant should be used. We revised our text throughout to clarify this point. We have previously shown that dominant negative ERL1DKinase override EPF1 peptide application (i.e. EPF1 peptide has no effects on epidermis of ERL1DKinase; Lee et al., 2012). Thus, we predict that ERL1DKinase would not be strongly influenced by the peptide application.

3) The authors use different markers to reveal the localization of ERL1-YFP. However, a typical experiment to study endocytosis is the co-localization with FM4-64. Figure 2—figure supplement 2A and Figure 3E provide some idea, but the individual endosomes should be tracked. BFA bodies and MVBs are a proxy to study this, but then the phrasing in the text should be accordingly.

This is a great suggestion. We observed co-localized FM4-64 endosomes and ERL1-YFP signals (now provided in revised Figure 1A), which indeed serves as a starting point to investigate the subcellular dynamics of ERL1. It is evident from our Concanamycin A treatment (revised Figure 2) that FM4-64 signals are eventually detected to a tonoplast and ERL1-YFP targeted into a vacuole. Having said that, in our hands it was not trivial to track the exact snapshots of ERL1 subcellular loalizations in a time-resolved fashion by simply following FM4-64 signals in different endomembrane organelles (e.g. early endosomes, late endosomes, etc). We strongly believe that the use of well-studied endomembrane organelle markers combined with pharmacological analysis provide robust and reliable results. These markers and inhibitors are widely used in the field.

4) TMM is required for endocytosis of ERL1, and not recycling. BFA blocks the secretion of ERL1. One would expect that if there is less endocytosis, there is also less ERL1 in BFA bodies. However, based on the data in Figure 2 this is not the case. Also, Figure 2B shows an increased ERL1-YFP intensity upon BFA treatment in tmm, so both the number of cells with BFA bodies and the signal intensity would be meaningful to report. I would rephrase the conclusion in subsection “TMM is required for the process of ERL1 endocytosis in true leaves”: the absence of tmm does not impact the ERL1 secretory pathway, as reflected in the presence of ERL1-YFP positive BFA bodies.

Both reviewer 1 and reviewer 2 also commented on this. As our response to other reviewers, we have re-analyzed the effects of BFA on ERL1-YFP in WT vs. *tmm* backgrounds by quantifying the ratio of YFP+ signals within the cell (voxel) per YFP+ signals at plasma membrane using our maximal intensity projection of Z-stack images (formerly Figure 2C, now Figure 4D). The analysis revealed a statistically significant reduction of ERL1-YFP in BFA bodies in *tmm* mutant. As reviewer 3 thoughtful points out, we revised our Results section based on the new analysis. Thank you.

5) The authors invoke that distinct EPF/EPFL peptide ligands activate a sub-population of ERL1 receptor complex to internalize through a TMM-based discriminatory system. This seems independent of distinct expression domains of these peptides as purified EPFL6 does not have the same effect as EPF1. The authors suggest distinct localization in nanodomains, but do not investigate this. It would be valuable to see FRAP analyses on ERL1-YFP +/1 EPF1 or EPFL6 in erl1 and erl1 tmm.

This is a very interesting point, and certainly it is our future direction to pursue specific nanodomain formation, using FRAP and elaborated microscopy, such as VAEM (as encouraged by reviewer 2).

6) Throughout the manuscript, MVBs should be used when describing the "large dots" on the figures. I am not sure if the resolution allows to see endosomes. Are Figure 4E-F and Figure 5E-F really showing number of endosomes or should this be MVBs?

As suggested, we have modified to MVP/LE, which is also part of late endosome.

7) The authors write "TMM is required for endocytic sorting of ERL1 to MVBs, which is a hallmark for receptor degradation". I guess that given the fact that in tmm ERL1-YFP signal increases in the ER, it is rather difficult to investigate this statement using e.g. Western Blot.

We agree that our original statement might have been too strong. The sentence was removed.

As requested by reviewer 2, we have done pharmacological treatment of ERL1-

YFP seedlings with Concanamycin A, which specifically inhibits vacuolar H^+^ATPase and thus reduce protein degradation in the lytic vacuole. We followed the published protocols (e.g. Kleine-Vehn et al., 2008) and pre-treated with FM464 to track a tonoplast. Our result show that ERL1 indeed accumulates in a vacuole in the presence of Concanamycin A (see revised Figure 2), indicating that ERL1 is targeted to a vacuole. We discussed receptor degradation in the context of our new results.

8) Given the recent knowledge on Tyr A23, I would only include the more specific ES9-17. Is this effect on ES9-17 specific for ERL1-YFP, or are other receptor proteins also accumulating in the ER?

ES9-17 is very new, and as far as we believe we are one of the very first to use this inhibitor in the context of shoot stomatal meristemoids. Although TyrA23 inhibits endocytosis is due to acidification, thereby non-specific manner (Dejonghe, 2016), it has been widely used to block internalization.

Indeed, since the publication of Dejonghe paper, it is still used to study endocytosis of LRR-RLKs (e.g. Song et al., 2019). While we agree with reviewer 3 that ES9-17's effects on endocytosis is more specific and thus clean, we strongly believe that presenting our data that both TyrA23 and ES9-17 triggers the ER accumulation of ERL1-YFP would strengthen our conclusion. To address the specificity to ERL1-YFP, we think that testing Stomage application to other LRR-RLK expressed endogenously in the leaf epidermis would be the better control. To this end, as also suggested by reviewer 2, we examined the effects of Stomagen peptide on the subcellular localization of BRI1 using a published

*BRI1pro::BRI1-GFP* reporter line (Friedrichsen et al., 2000). The Stomagen application did not change plasma-membrane localization of BRI1, thus the influence of Stomagen is likely specific to ERL1-YFP. Please see Figure 8—figure supplement 3 or our revision.

[Editors’ note: what follows is the authors’ response to the second round of review.]

Your manuscript reveals remarkable insights into peptide ligands and how they modulate the intracellular distribution of their receptors. We invite you to resubmit your paper for final acceptance, considering the following main suggestions of the reviewers:

We truly thank for the positive and encouraging comments.

I) The reviewers ask you to clarify several issues, such as statistical evaluation and quantifications, which may not require novel experiments.

As requested by reviewer 1, we have in addition performed two-way ANOVA for analysis for those data with two parameters, treatment and genotype (Figure 4D, 4F, Figure 4—figure supplement 1, Figure 4—figure supplement 2D, Figure 5-Figure supplement 1B, 1C in the revision). Accordingly, the additional R scripts for the analysis are provided as Source code.

II) During their discussion, the reviewers defined a better characterization and integration of the ERL1Δkinase line, as an essential part of the revision, and one that unfortunately requires additional experimentation.

We agree that testing how ERL1ΔK responds to different ligands is very important to complete our story. As you might hopefully recognize, however, this has been extremely challenging for us. The Torii laboratory at the University of Texas at Austin has been under lock down due to COVID-19 surge in the US, notably terrible in the Southern states including Texas. Even worse, we ran out of bioactive recombinant MEPF1 peptide.

After UT-Austin gave us permission to conduct time-sensitive experiments in July, my postdoc, Dr. Pengfei Bai, who is now a co-author, worked day and night to express, purify, refold, and test bioactivities of their recombinant MEPF1 peptide. The bioactive peptide was then shipped to Japan, where Dr. Akira Yoshinari performed the requested experiments.

As you can see in revised Figure 8, neither MEPF1 nor Stomagen peptide application triggered subcellular localization changes of the dominant-negative ERL1ΔK. ERL1ΔK remained predominantly at the plasma membrane upon MEPF1 or Stomagen application. These new findings suggest that ERL1ΔK is dysfunctional in ligand-induced subcellular trafficking. Because ERL1ΔK lacks the entire cytoplasmic kinase domain, including juxtamembrane domain, kinase domain, and C-terminal tail region, we predict that proper interaction site with endocytosis cargo adaptors may have been lost in ERL1ΔK. While identifying specific cargo adaptors for ERL1 is a future study that is out-of-scope of this manuscript, our additional experiments highlights a highly-tuned, complex actions of antagonistic peptides on ERL1 subcellular dynamics is a highly tuned, coordinated process. We have revised our main text to include these new findings (subsection “Cytoplasmic domain of ERL1 is required for proper subcellular trafficking behavior”, Discussion section).

III) The reviewers judge that some experiments aimed at a better integration of your data into developmental context would substantially strengthen the impact of your manuscript. On the other hand, they do not believe that such an insight is essential for acceptance of the manuscript.

We appreciate the editors' advice. Please see our detailed response to reviewer 2 regarding this point.

Please, see the detailed comments of the reviewers below.Reviewer #2:The authors undertook a remarkable effort to improve the cell biological aspects of this manuscript and I do appreciate the new experiments and careful quantification of the BFA experiments and analysis of new marker lines with expert Dr. Akira Yoshinari. In addition, the ConcavA treatments and the clear visualization of ERL1 in vacuoles adds significantly to this manuscript. However, I would like to stress two of my previous concerns again.

We truly thank reviewer 2 for commending our "remarkable effort to improve the cell biological aspects of this manuscript".

1) Previous point 1:From a mechanistic/genetic perspective, the functional relevance of ERL1 endocytosis and targeting to MVB/LE and vacuole is still missing. I do completely understand that a phenotypic assay is not doable due to long-time cytotoxic effects of BFA or Wm. I also agree that strong genetic support can only come through analysis of how ERL1 is modified (phosphorylation etc) and how these modifications affect endocytosis, but this is clearly out of scope of this article. Nevertheless, in Qi et al., 2017, the authors show that the EPF1-ERL1 module directly represses MUTE in an autocrine manner. When EPF1 expression is induced, a clear decrease in MUTE-YFP fluorescence can be observed already 15hours post induction. I wonder whether induction of EPF1 together with mock or toxin treatment (Wm?) in the first couple of hours could reveal differences in how efficiently MUTE-YFP is downregulated. This could give at least somewhat functional support to the hypothesis that the endocytosis and targeting to the vacuole is required for efficient downregulation of key targets by the EFP1-ERL1 module.

We appreciate reviewer 2's comments. As we all know, chemical inhibitors of membrane trafficking, such as BFA, TyrA23, and Concanamycin A, are a very powerful tool to delineate subcellular dynamics of membrane proteins of interest, whether receptor kinases or transporters. However, there are major problems with the use of these chemical inhibitors for extended period of time to study developmental outcomes.

First and foremost, these inhibitors are not specific to one plasma membrane protein of interest, but rather affect many proteins. It is well known that BFA, for instance, affects recycling of PIN auxin efflux carriers (e.g. Kleine-Vehn et al., 2008), which also plays a role in stomatal patterning (e.g. Le et al., 2014). ES9-17 blocks endocytosis of not only ERL1, but also PIN2 (Iwatate et al., 2020), BRI1 (Dejonghe et al., 2019), and likely many other membrane proteins. Wm could somehow block endocytosis (Emans et al., 2002; Dettmer et al., 2006) in addition to vacuolar sorting. It is therefore impossible to conclude that a developmental phenotype caused by these inhibitors are due to trafficking defects of a single membrane protein. Second, as reviewer 2 acknowledges, these inhibitors possess high cellular toxicity. Extended treatment of seedlings with these chemicals will impact seedling growth and cell proliferation, making it difficult, if not impossible, to characterize stomatal development phenotype. We appreciate that reviewer 2 suggested using a reporter MUTE-YFP line as a quicker marker (~15 hours). However, in our experience, addition of these chemicals for only a few hours would impact the health of the seedlings, thus the interpretation of these experiments would be very tricky.

Our manuscript reports unique, subcellular dynamics of a single receptor (ERL1) by different peptide ligands with distinct biological activities, thereby suggesting the intricate, fine-tuning of developmental patterning by multiple peptide signals. Future analysis, such as identifying critical post-translational modifications of ERL1 and characterizing their impacts on receptor subcellular behavior, will provide a full picture of how intricate regulation of receptor dynamics directly specifies stomatal patterning.

2) Previous point 12:Even though the authors stated in their response that they elaborated on ER ERL1 and ERL2 redundancy in the discussion, I cannot find new information or statements regarding the fact that the Stomagen-induced ER-retainment of ERL1 is much stronger in the triple erecta-erl1-erl2 background. I am aware that this is a subtle difference but as the authors confirm an important one. So please elaborate on this in the results (line 298ff) and clearly mention that Stomagen has been shown to interact initially with ER before acting through ERL1 and ERL2 at later stages, which is why a triple mutant is needed here and ER retainment in the single erl1 is much weaker.

We fully agree with reviewer 2 that the Endoplasmic Reticulum retention of ERL1-YFP is notable in the *erecta erl1 erl2* mutant background, i.e. when functional *ERECTA* and *ERL2* genes are missing. As reviewer 2 astutely points out, we also think that the lack of ERECTA, the major receptor for the early stage of stomatal initiation, confers sensitized, exaggerated response of ERL1-YFP to Stomagen. This is because ERL1 is the only receptor of the family to perceive Stomagen. In addition, it is known that the lack of ERECTA triggers excessive entry divisions of stomatal cell lineages and clustering of meristemoids, where ERL1 is expressed (e.g. Shpak et al., 2005, Lee et al., 2012, and Qi et al., 2017). The dysregulated cell-cell signaling in the *erecta erl1 erl2* background likely abrogate the 'buffering' of exogenously applied Stomagen peptide.

In response to reviewer 2, we have elaborated on the main text, stating the following sentences. We sincerely thank reviewer 2 for the keen, thoughtful suggestion.

"The endoplasmic reticulum retention of ERL1-YFP is notable in the *erecta erl1 erl2* mutant background, i.e. when functional *ERECTA* and *ERL2* genes are missing. The signals in *erl1* ("WT") are much weaker. Because Stomagen interacts with ERECTA at the early stage of stomatal initiation, the absence of *ERECTA* likely confers sensitized, exaggerated response of ERL1-YFP to Stomagen. Alternatively, a dysregulated cell-cell signaling in the *erecta erl1 erl2* background may abrogate the buffering of exogenously applied Stomagen peptide." (Discussion section.)